# WHY DPO IS A MISSPECIFIED ESTIMATOR AND HOW TO FIX IT

**Aditya Gopalan**
Indian Institute of Science
Bangalore, India
*aditya@iisc.ac.in*

**Sayak Ray Chowdhury**
Indian Institute of Technology
Kanpur, India
*sayakrc@iitk.ac.in*

**Debangshu Banerjee**
HP AI Research
Bangalore, India
*debangshu.banerjee@hp.com*

## ABSTRACT

Direct alignment algorithms such as Direct Preference Optimization (DPO) fine-tune models based on preference data, using only supervised learning instead of two-stage reinforcement learning with human feedback (RLHF). We show that DPO encodes a statistical estimation problem over reward functions induced by a parametric policy class. When the true reward function that generates preferences cannot be realized via the policy class, DPO becomes misspecified, resulting in failure modes such as preference order reversal, worsening of policy reward, and high sensitivity to the input preference data distribution. On the other hand, we study the local behavior of two-stage RLHF for a parametric class and relate it to a natural gradient step in policy space. Our fine-grained geometric characterization allows us to propose AuxDPO, which introduces additional auxiliary variables in the DPO loss function to help move towards the RLHF solution in a principled manner and mitigate the misspecification in DPO. We empirically demonstrate the superior performance of AuxDPO on didactic bandit settings as well as LLM alignment tasks.

## 1 INTRODUCTION

Preference-based alignment is a key part of the training process of large language models (LLMs). It aims to steer a pretrained model's conditional distribution toward outputs that humans (or carefully calibrated annotator models) prefer. Formally, given comparison data $(s, a_w, a_l)$, the goal is to shape a policy $\pi$ whose induced responses align with a latent reward model that generated those preferences.

Two-stage RLHF is the standard way of carrying out preference-based alignment (Ziegler et al., 2019). However, it is computationally demanding (it requires training a separate reward model) and complex due to a two-stage pipeline (supervised learning for the reward model followed by RL policy optimization based on the learned reward model). Concretely, the reward model $r_\phi(s, a)$ is trained on preference pairs via a Bradley–Terry/Logistic objective (Bradley and Terry, 1952), maximizing $\log \sigma(r_\phi(s, a_w) - r_\phi(s, a_l))$ over $(s, a_w, a_l)$. The second stage then optimizes a KL-regularized objective of the form $\max_\pi \; \mathbb{E}_{s \sim \rho, \, a \sim \pi(\cdot|s)}[r_\phi(s, a)] \; - \; \beta \, D_{\mathrm{KL}}(\pi(\cdot \mid s) \, \| \, \pi_{\mathrm{ref}}(\cdot \mid s))$, typically implemented with PPO-style updates. This stage is on-policy and rollout-heavy: the model must repeatedly generate samples to estimate advantages under $r_\phi$, maintain a stable KL to the reference policy $\pi_{\mathrm{ref}}$ (often the SFT model), and tune sensitive hyperparameters (e.g., $\beta$, clip ranges, learning rates). In practice, this entails nontrivial engineering (reward hacking mitigation, variance reduction, response-length control) and significant compute for both reward-model training and RL updates, which motivates interest in lighter-weight alternatives.

The introduction of direct alignment algorithms such as Direct Preference Optimization (DPO) (Rafailov et al., 2023) was a landmark step that paved the way for lightweight alignment of a base model using preference data and only a single supervised training phase. DPO operates by explicitly solving the second, KL-regularized, policy optimization phase of RLHF and using it to reparameterize the first phase of reward learning in terms of the optimized policy, in effect achieving a one-step equivalent to the original two-step pipeline. This has been instrumental in enabling both industrial players and the open-source AI community to carry out fast alignment of models without the burden

of additional resources. Many variants of DPO have since been developed catering to various aspects of direct alignment.

Despite its widespread appeal, however, the design of DPO rests on the idealized assumption that the policy class is *tabular*, i.e., it includes every possible input-output conditional probability distribution $(\pi(a \mid s))_{s,a}$, where $s$ and $a$ denote prompt and response strings, respectively. This assumption enables the KL-regularized policy optimization problem to be solved in closed form and used explicitly to derive the equivalent supervised DPO loss (Rafailov et al., 2023, Appendix A.1).

In contrast, real-world LLMs are far from tabular and are, in fact, parametric policy classes, resulting naturally from the use of neural architectures (e.g., Transformers) with only a finite number of parameters. One may then ask: does minimizing the DPO loss over a non-tabular policy class still preserve the claimed equivalence with full two-stage RLHF? If not, then how does it differ from the ideal RLHF-optimal policy? Does it enjoy any guarantees with respect to the performance of the latter, and, if not, is there a principled fix?

We address these questions by introducing a systematic framework to uncover the geometry of direct preference optimization in parametric policy classes. Our study helps show how DPO essentially solves a misspecified statistical estimation problem in the space of reward functions that are implicitly parameterized by the underlying policy class. Misspecified estimation problems have been analyzed, in the statistics literature, for exhibiting undesirable phenomena such as inconsistent and arbitrary estimates that are sensitive to the input data distribution (White, 1982); we show that such phenomena also manifest in the DPO setting. Our analysis framework also allows us to modify DPO in a principled manner, to yield a new algorithm (AuxDPO) towards achieving the performance of two-stage RLHF in parametric models. More specifically, we make the following contributions:

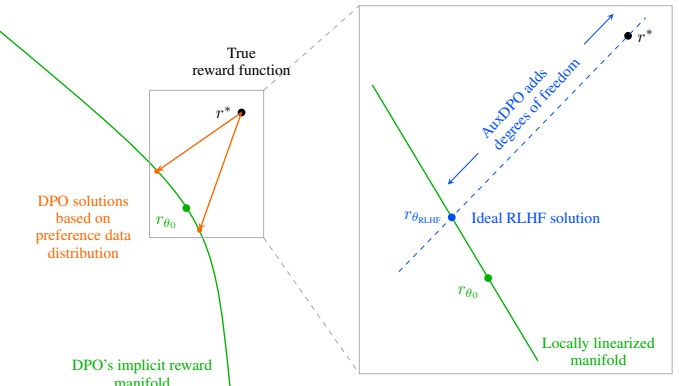

Figure 1: The geometry of DPO for parametric policies. (Left) DPO essentially performs a projection of the true preference-generating reward function ($r^*$ in black) onto the manifold of reward functions implicitly expressed by the policy class. If $r^*$ is in the manifold, then DPO finds the correct KL-regularized RLHF policy, but otherwise, the policy found (any orange point) is unreliable. (Right, zoomed inset) Locally linearizing the manifold around the base policy's implicit reward function ($r_{\theta_0}$) uncovers geometric insights. To reliably drive the solution to the reward function corresponding to the ideal RLHF solution ($r_{\theta_{\mathrm{RLHF}}}$ in blue), AuxDPO introduces additional controlled degrees of freedom, along the null space of a base-policy dependent matrix to sidestep misspecification.

1. We show that for general parametric policy classes, there is a misspecified statistical estimation problem at the core of the DPO algorithm by design: DPO loss minimization is equivalent to a weighted KL-projection of the true reward function $r^*$ onto the (parametric, lower-dimensional) manifold of reward functions induced by the policy class. The weights of the projection are governed by the preference data collection frequencies (Fig. 1, left).

2. We show that DPO, in the misspecified setting, can suffer from various failure modes such as order reversal of preferences, overall reward reduction, sensitivity to preference data frequencies, etc. These failure modes occur even with 'clean data', i.e., infinite preference data generated using a BTL model based on an underlying true reward function $r^*$ and fed to DPO. Our analysis is based on taking a local, linearized view of DPO's implicit reward function manifold, which is accurate in the large-$\beta$ regime.

3. On the other hand, studying the local geometry of two-stage RLHF for general parametric policy classes yields new insights about linear equivalence classes of reward functions. We use these insights to design AuxDPO, a new direct preference optimization algorithm that effectively mitigates the misspecification issue by introducing auxiliary controlled degress of freedom in reward space (Fig. 1, right). We demonstrate the effectiveness of AuxDPO in experiments. On

real-world LLM preference tuning tasks, AuxDPO consistently outperforms DPO in aligning to held-out human preferences, confirming its practical value.

**Related work.** A recent line of work focuses on studying the insufficiency and implications of the tabular policy class assumption. Gao et al. (2024) and Swamy et al. (2025) call into question the tabular policy class assumption in the context of original two-stage RLHF. Tajwar et al. (2024) carry out an empirical investigation and note that the standard DPO loss can inadvertently reduce the model's absolute likelihood of chosen responses as long as the relative probability between chosen and rejected responses increases. Meng et al. (2024) and Xu et al. (2024a) propose fixes based on considerations of margin and length normalization, and elimination of the reference policy.

Xu et al. (2024b) and Song et al. (2024) study the shortcomings of DPO arising from a lack of coverage, arguing that DPO can fail if a strong coverage condition is not met. The latter provide a counterexample to this end, showing the existence of an implicitly expressible reward function that is $\varepsilon$-approximately close to the true reward function but corresponds to a policy not in the KL neighborhood of the base policy. It is, however, unclear if such an implicit reward function can actually be output by DPO. Our fine-grained analysis in this paper shows that even with perfect coverage (uniform base policy), the policy returned by DPO can suffer from pathologies such as preference reordering and a decrease of overall expected reward (Proposition 3).

A separate line of work focuses on the gradient dynamics of DPO loss optimization and its impact on policy probabilities (Pal et al., 2024; Razin et al., 2024; Jian et al., 2025). It is shown that an individual gradient step on the standard DPO loss can result in likelihood displacement, where the probability of preferred responses can drop relative to the base policy for a gradient step. Our approach eschews assuming any specific optimization algorithm such as gradient descent and considering individual gradient steps, and instead focuses on showing failure modes such as likelihood displacement, preference reversal, reward reduction, etc. by studying the minimizer of the DPO loss.

Shi et al. (2025), perhaps the closest in spirit to our study, demonstrate a multi-armed bandit example that, when subjected to DPO with a log-linear policy class, does not cause any movement from the base policy. However, this example relies on a degenerate and symmetric reward function; we are able to demonstrate, in a fine-grained manner, that the policy can strictly worsen from the base policy in a manner that is highly sensitive to the preference data distribution (Proposition 3).

## 2 PRELIMINARIES

Let $\mathcal{D} = \left\{ \left( s^{(i)}, a_w^{(i)}, a_l^{(i)} \right) : i \in [n] \right\}$ be a dataset of $n$ samples, where each sample has a prompt $s \in \mathcal{S}$, two responses $a_w, a_l \in \mathcal{A}$ such that $a_w \succ a_l$, i.e., $a_w$ is a preferred response over $a_l$. We assume both $\mathcal{S}$ and $\mathcal{A}$ are finite sets, with $|\mathcal{S}| \cdot |\mathcal{A}| = m$. The prompt $s$ is sampled from a distribution $\rho$ over $\mathcal{S}$. The pair of responses $(a_w, a_l)$ is sampled from some base (reference) policy $\pi_{\text{ref}}$ conditioned on $s$, i.e., $a_w, a_l \sim \pi_{\text{ref}}(\cdot|s)$. The preference ordering between a pair of responses is assumed to be sampled according to a Bradley-Terry-Luce (BTL) model: the probability of $a$ being preferred to $a'$ is given by $p_{s,a,a'}^{\text{BTL}}(r^*) = \sigma(r^*(s, a) - r^*(s, a'))$, where $r^* : \mathcal{S} \times \mathcal{A} \to \mathbb{R}$ is a (latent) reward function and $\sigma(z) := \frac{1}{1+e^{-z}}$ is the sigmoid function.

Let $\pi_\theta : \mathcal{S} \to \Delta(\mathcal{A})$ be a policy (e.g., a language model) smoothly parameterized by a $d$-dimensional vector $\theta \in \mathbb{R}^d$ (e.g., the weights of a transformer), where $\Delta(\mathcal{A})$ denotes the probability simplex over $\mathcal{A}$. Let $\theta_0 \in \mathbb{R}^d$ be the parameter for the base policy $\pi_{\text{ref}}$ so that $\pi_{\text{ref}} = \pi_{\theta_0}$. A special case is the tabular policy class, where $d = m = |S| \cdot |A|$ and $\pi_\theta(a|s) = \theta_{s,a}$ (assuming without loss of generality that $\sum_{s,a} \theta_{s,a} = 1$). However, LLM policy classes are structured and non-tabular with parameter dimension $d \ll m$, e.g., the neural softmax policy $\pi_\theta(a \mid s) = \frac{\exp(f_\theta(s,a))}{\sum_{a' \in \mathcal{A}} \exp(f_\theta(s,a'))}$, where $f_\theta$ is, say, a neural network.

For a given reward function $r^*$, the optimal policy in a KL-regularized sense is obtained by maximizing the following objective:

$$J(\theta; r^*) = \mathbb{E}_{\rho, \pi_\theta} \left[ r^*(s, a) - \beta \log \frac{\pi_\theta(a|s)}{\pi_{\theta_0}(a|s)} \right] = \mathbb{E}_{\rho, \pi_\theta} \left[ r^*(s, a) - \beta D_{\text{KL}}(\pi_\theta(\cdot|s) || \pi_{\theta_0}(\cdot|s)) \right], \quad (1)$$

where $\mathbb{E}_{\rho,\pi_\theta}$ denotes expectation taken over $s \sim \rho(\cdot)$ and $a \sim \pi_\theta(\cdot \mid s)$, and $\beta > 0$ is a parameter that controls the amount of deviation from the base policy. We assume that $\theta^* \in \mathbb{R}^d$ is the unique minimizer of (1). For our analytical results, we will focus on the 'local' case $\beta \gg 1$, meaning that the policy is not allowed to move beyond a local neighborhood of $\pi_{\theta_0}$. When the policy class is tabular, it follows that the optimal policy $\pi_{\theta^*}$ and the latent reward $r^*$ satisfy

$$\pi_{\theta^*}(a|s) = \frac{1}{Z^*(s)}\pi_{\theta_0}(a|s)\exp(r^*(s,a)/\beta) \iff r^*(s,a) = \beta\log\frac{\pi_{\theta^*}(a|s)}{\pi_{\theta_0}(a|s)} + \beta\log Z^*(s) \ , \quad (2)$$

where $Z^*(s) = \sum_{a \in \mathcal{A}} \pi_{\theta_0}(a|s)\exp(r^*(s,a)/\beta)$ is the normalizing or partition function (Rafailov et al., 2023). Under this reward-policy equivalence, the preference probabilities under the BTL model can be expressed using the optimal policy $\pi_{\theta^*}$ and the base policy $\pi_{\theta_0}$ as follows.

$$p_{s,a,a'}^{\mathrm{BTL}}(r^*) = \sigma\left(\beta\log\frac{\pi_{\theta^*}(a|s)}{\pi_{\theta_0}(a|s)} - \beta\log\frac{\pi_{\theta^*}(a'|s)}{\pi_{\theta_0}(a'|s)}\right) = \sigma\left(r_{\theta^*}^\beta(s,a) - r_{\theta^*}^\beta(s,a')\right) \ ,$$

where, for any $\theta \in \Theta$, $r_\theta^\beta : \mathcal{S} \times \mathcal{A} \to \mathbb{R}$ defined via $r_\theta^\beta(s,a) := \beta\log\frac{\pi_\theta(a|s)}{\pi_{\theta_0}(a|s)}$ denotes the implicit reward function corresponding to the policy $\pi_\theta$ at deviation level $\beta$ (note that $r_{\theta_0}^\beta \equiv 0$ by definition). Let $\mathcal{R}^\beta = \left\{ r_\theta^\beta : \theta \in \mathbb{R}^d \right\} \subsetneq \mathbb{R}^m$ be the set of all implicit reward functions induced by the policy parameters $\theta$ at deviation level $\beta$. Given the dataset $\mathcal{D}$, DPO (Rafailov et al., 2023) finds the minimizer of the *empirical* DPO loss

$$\mathcal{L}_{\mathcal{D}}(\theta) = -\sum_{i=1}^{n}\log\sigma\left(r_\theta^\beta(s^{(i)}, a_w^{(i)}) - r_\theta^\beta(s^{(i)}, a_l^{(i)})\right) = -\sum_{s,a_w,a_l} N_{s,a_w,a_l}\log\sigma\left(r_\theta^\beta(s,a_w) - r_\theta^\beta(s,a_l)\right),$$

where $N_{s,a_w,a_l}$ is the total number of pairwise preferences for which $a_w \succ a_l$ at $s$. If $n_{s,a,a'}$ denotes the total number of pairwise preferences for the triplet $(s,a,a')$ (assumed non-random and fixed in advance), then $N_{s,a_w,a_l} \sim \texttt{Binomial}\left(n_{s,a_w,a_l}, p_{s,a_w,a_l}^{\mathrm{BTL}}(r^*)\right)$, yielding the *population* DPO loss

$$\mathcal{L}(\theta) = -\sum_{s,a,a'} n_{s,a,a'}\left[p_{s,a,a'}^{\mathrm{BTL}}(r^*)\log p_{s,a,a'}^{\mathrm{BTL}}(r_\theta^\beta) + (1 - p_{s,a,a'}^{\mathrm{BTL}}(r^*))\log\left(1 - p_{s,a,a'}^{\mathrm{BTL}}(r_\theta^\beta)\right)\right]. \quad (3)$$

We take up this loss for DPO as our main object of study in the sequel.

## 3 Reward Misspecification in DPO

Let $r^*$ denote the $m$-dimensional vector of latent rewards $(r^*(s,a))_{s,a}$, by slightly abusing notation.

**Proposition 1** (DPO is weighted KL-projection). *Assume that the pairwise preference data are drawn from $p_{s,a_w,a_l}^{\mathrm{BTL}}(r^*)$ for some $r^* \in \mathbb{R}^m$, with $n_{s,a,a'}$ preference pairs drawn for each triplet $(s,a,a')$. If $\theta_{DPO}$ minimizes the DPO loss (3), then its corresponding implicit reward function satisfies*

$$r_{\theta_{DPO}}^\beta = \arg\min_{r \in \mathcal{R}^\beta} \sum_{s,a,a'} n_{s,a,a'}\, d_{KL}\left(p_{s,a,a'}^{\mathrm{BTL}}(r^*) \| p_{s,a,a'}^{\mathrm{BTL}}(r)\right) , \quad (4)$$

*where $d_{KL}(p\|q)$ is the Kullback-Leibler divergence between two Bernoulli random variables with parameters $p, q$.*

The result establishes that DPO projects (according to reverse-KL divergence weighted by pairwise preference counts $n_{s,a,a'}$) the true reward function $r^*$ onto the set of implicit reward functions $\mathcal{R}^\beta$ (with the corresponding policy being returned after fine-tuning). It implies that if $r^*$ is realizable, i.e., $r^* = r_\theta^\beta$ for some $\theta$, then this projection (trivially) finds $r_\theta^\beta$ and hence the policy $\pi_\theta$, which coincides with the RLHF policy, i.e., $\theta = \theta^*$.

However, if $r^* \notin \mathcal{R}^\beta$ (which is typically the case since $\mathcal{R}^\beta$ is a lower-dimensional ($d$-dimensional) manifold of $\mathbb{R}^m$), then we are in the misspecified estimation setting. In this case, the result of the KL-projection will, in general, be dependent on the exact weighted projection which is determined by the preference data frequencies $(n_{s,a,a'})_{s,a,a'}$. We demonstrate, in the next section, that the policies resulting from DPO's misspecified estimation enjoy no guarantees: they could suffer from preference reversal, or even worse, yield a lower average reward than even the base policy (contrary to two-stage RLHF, where the average reward can never decrease).

## 3.1 LOCAL GEOMETRY OF DPO

Let us locally approximate the implicit reward $r_\theta^\beta(s, a)$ via its first-order Taylor expansion around $\theta_0$:

$$r_\theta^\beta(s, a) \approx r_{\theta_0}^\beta(s, a) + \langle \nabla r_{\theta_0}(s, a), \theta - \theta_0 \rangle = \beta \langle \tfrac{\nabla \pi_{\theta_0}(a|s)}{\pi_{\theta_0}(a|s)}, \theta - \theta_0 \rangle = \beta \nabla \log \pi_{\theta_0}(a|s)^\top (\theta - \theta_0).$$

Define the $d \times m$ Jacobian matrix $A_{\theta_0}$ as the matrix containing $\nabla \log \pi_{\theta_0}(a|s)$ in its $(s, a)$-th column. With this, the local linear approximation of $r_\theta^\beta$ takes the form $\bar{r}_\theta^\beta = \beta A_{\theta_0}^\top (\theta - \theta_0)$. Since $\{\bar{r}_\theta^\beta : \theta \in \mathbb{R}^d\}$ is the column space of $A_\theta^\top$, we arrive at the linear manifold approximation $\mathcal{R}^\beta \approx \mathcal{C}(A_{\theta_0}^\top)$, for $\theta$ in the local neighborhood of $\theta_0$. Note that this linear approximation is independent of the choice of $\beta$ and is a function only of the policy class and base policy $\pi_{\theta_0}$.

**Remark 2.** *The error in the linear approximation of the manifold $\mathcal{R}^\beta$ by $\mathcal{C}(A_{\theta_0}^\top)$ can be controlled to within any arbitrary tolerance by taking the policy deviation parameter $\beta$ to be sufficiently large. Proposition 8 formally controls the approximation error. In the sequel, we only work with the linear approximation to develop our results.*

Armed with this local linearization of the implicit reward manifold, we now show an example of a single-prompt, 3-response setting, with a 1-dimensional policy parameter, in which DPO exhibits counterintuitive and unexpected behavior, including preference reversal, reward reduction, and high sensitivity to the pairwise preference data counts $n_{s,a,a'}$.

**Proposition 3** (Example of DPO with preference reversal and reward decrease). *There exists a promptless policy optimization problem with three responses and linear softmax policy class parameterized with a 1-dimensional parameter $\theta$ and a true reward vector $r^* \in \mathbb{R}^3$ such that DPO, carried out with pairwise preferences generated according to BTL($r^*$), sufficiently large $\beta$ and pairwise counts $\{n_{i,j}\}$, yields a policy $\pi_\theta$ such that (i) $\pi_\theta$ favors the response with second highest reward, (ii) $\pi_\theta$ decreases (resp. increases) the probability of the action with the highest reward (resp. second highest reward) with respect to the base policy $\pi_{\theta_0}$, and (iii) $\pi_\theta^\top r^* < \pi_{\theta_0}^\top r^*$.*

*Proof.* Consider the following example with 3 responses $a_1, a_2, a_3$ with latent rewards $r^* = [2, 3, 1]$, which yields the preference order $a_2 \succ a_1 \succ a_3$. The pairwise preference counts in the dataset are highly imbalanced in a way that $n_{3,1} \gg \max\{n_{1,2}, n_{2,3}\}$. The policy is given by $\pi_\theta = \frac{1}{Z}[e^\theta, e^{-\theta}, 1]$, where $Z = 1 + e^\theta + e^{-\theta}$. We take a uniform base policy (i.e., $\theta_0 = 0$). Under the BTL model, $r^* \equiv [1, 2, 0]$ since both induce the same preference distribution. Hence, we will work with this equivalent $r^*$. The setting is depicted in Figure 2.

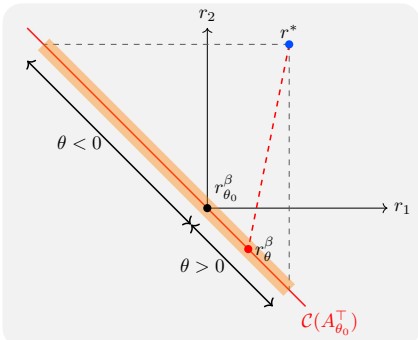

In this setting, the policy gradient matrix takes the form $A_\theta = \frac{1}{Z}[1 + 2e^{-\theta}, -(1 + 2e^\theta), e^{-\theta} - e^\theta]$. Hence $A_{\theta_0} = [1, -1, 0]$ and thus $\mathcal{R}^\beta \approx \text{span}([1, -1, 0])$ in the local neighborhood of $\theta_0 = 0$. Now, since $n_{1,3}$ dominates the other two comparison counts, the solution to the optimization problem in (4), e.g.,

$$\arg\min_{r \in \mathcal{R}^\beta} \sum_{i \neq j} n_{i,j} d_{\text{KL}}\left(p_{i,j}^{\text{BTL}}(r^*) \| p_{i,j}^{\text{BTL}}(r)\right)$$

will push the probability that $a_1$ is preferred over $a_3$, i.e., $p_{1,3}^{\text{BTL}}(r_\theta^\beta)$ close to $p_{1,3}^{\text{BTL}}(r^*)$, yielding $r_\theta^\beta(a_1) - r_\theta^\beta(a_3) = O(\alpha)$ for some $\alpha > 0$. Moreover, since $r_\theta^\beta$ should lie in $\text{span}([1, -1, 0])$, DPO will end up with the reward function $r_\theta^\beta \approx [\alpha, -\alpha, 0]$. This, in turn, will make $a_1$ to be preferred over $a_2$ by the learned reward, indicating a preference reversal from that given by $r^*$. Furthermore, from the equivalence relation (5), we get the post-optimized policy parameter $\theta = O(\alpha)$. This yields (i) $\pi_\theta(a_1) > \pi_\theta(a_2)$ (the policy favors a suboptimal response) (ii) $\pi_\theta(a_2) < \pi_{\theta_0}(a_2)$ and $\pi_\theta(a_1) > \pi_{\theta_0}(a_1)$ (likelihood of the optimal response

Figure 2: An example with 3 responses and 1-d policy parameter showing failure modes of DPO. $r^*$ is the latent reward. The red line denotes the linear approximation $\mathcal{C}(A_{\theta_0}^\top)$ of the implicit reward manifold $\mathcal{R}^\beta$. The region shaded in orange represents all possible implicit reward functions that DPO can possibly project onto, depending on the relative proportion of pairwise preference counts $n_{1,2}, n_{2,3}, n_{3,1}$. If $n_{3,1}$ dominates the rest, then the projection $r_\theta^\beta$ induces a post-optimized policy parameter $\theta > 0$, leading to preference reversal and reduction of expected reward, causing DPO to fail.

decreases and that of a suboptimal response increases) and (iii) $\pi_\theta^\top r^* = \frac{e^\alpha + 2e^{-\alpha}}{1+e^\alpha+e^{-\alpha}} < 1 = \pi_{\theta_0}^\top r^*$ (average reward decreases relative to the base policy).

The example exhibits the following aspects:

1. There are pairs $(i,j)$ (e.g., $i = 2, j = 1$) where $a_i \succ a_j$ with respect to $r^*$ (i.e., $r^*(a_i) > r^*(a_j)$) but $\pi_\theta(a_i)$ decreases and $\pi_\theta(a_j)$ increases with respect to the base policy. This is more extreme than likelihood displacement, where $\pi_\theta(a_i)$ and $\pi_\theta(a_j)$ both decrease or increase together but their difference $\pi_\theta(a_i) - \pi_\theta(a_j)$ is presumed to increase (Razin et al., 2024; Pal et al., 2024).

2. The expected reward with respect to $r^*$ *decreases* from $\pi_{\theta_0}$, whereas two-stage RLHF would have increased the expected reward on any policy class (assuming $r^*$ is learnt accurately in the first stage).

3. Our example is based on the DPO population loss, which is effectively DPO operating in the data-rich regime where unlimited pairwise preference data from a BTL model are used as input. This circumvents the issue of failure modes of DPO known to occur due to scarce data sampling (Tajwar et al., 2024). We show that failure modes arise due to the inherent misspecified geometry induced by the lack of model capacity, interacting with the frequencies of pairwise sampling.

4. Our example applies in the strongest possible 'oracle' optimization model where we assume that the (population) DPO loss can be optimized *exactly*. Our results are not dependent on the idiosyncrasies or specifics of what algorithm is used to optimize the DPO loss (e.g., gradient descent and variants as explicitly considered in other works (Razin et al., 2024; Pal et al., 2024)), as long as it is (near) optimal.

5. Sensitivity of DPO with respect to the preference data distribution: In the example, if the pairwise preference counts are such that $n_{1,2} \gg \{n_{2,3}, n_{3,1}\}$, then DPO would learn the desired reward function $r_\theta^\beta \approx [-\alpha, \alpha, 0]$, which would, in turn, induce a policy parameter $\theta < 0$, escaping the failure modes effectively. Therefore, depending on the relative proportions of pairwise preference counts $\{n_{i,j}\}_{i,j}$, one could either get the desired result or the opposite one, as in the example, or perhaps no movement at all, making DPO sensitive to preference data sampling distribution. This sensitivity to the exact preference distribution also represents a failure mode of DPO.

**Remark 4** (Global coverage is not sufficient for optimality). *Song et al. (2024) argue that global coverage, i.e.,* $\max_{s,a} \frac{\pi_\theta(a|s)}{\pi_{\theta_0}(a|s)} \leqslant C \,\forall \theta \in \mathbb{R}^d$, is *necessary for DPO to converge to the optimal policy* $\theta^*$. *In our example,* $\pi_{\theta_0}$ *satisfies this with* $C = 3$. *However, this condition is* not sufficient *for DPO to perform optimally, since, as we have seen, DPO can learn an unaligned reward model depending on the relative frequency of preference counts.*

## 4 TOWARDS MITIGATING DPO'S PITFALLS

We propose to address the misspecification issue of DPO by first studying the nature of the RLHF policy optimization step in a suitably local sense (assuming ideal reward learning), and then using the insights gained to encourage movement towards this solution.

### 4.1 LOCAL GEOMETRY OF RLHF OPTIMIZATION

We locally approximate the objective $J(\theta; r^*)$ in (1) around the base policy $\pi_{\theta_0}$. To do so, we approximate the expected reward using a first-order Taylor series expansion and the KL penalty using a second-order Taylor series expansion:

$$\mathbb{E}_{\rho,\pi_\theta}[r^*(s,a)] \approx \mathbb{E}_{\rho,\pi_{\theta_0}}[r^*(s,a)] + (\theta-\theta_0)^\top A_{\theta_0} D_{\rho,\theta_0} r^*, \quad D_{\mathrm{KL}}(\pi_\theta || \pi_{\theta_0}) \approx \frac{1}{2}(\theta-\theta_0)^\top F_{\rho,\theta_0}(\theta-\theta_0),$$

where $D_{\rho,\theta_0}$ is a diagonal matrix with scaled base policy-probabilities $\rho(s)\pi_{\theta_0}(a|s)$ in the diagonal entries, and $F_{\rho,\theta_0} = \mathbb{E}_{\rho,\pi_{\theta_0}}\left[\nabla \log \pi_{\theta_0}(a|s)\nabla \log \pi_{\theta_0}(a|s)^\top\right]$ denotes the Fisher information matrix at $\pi_{\theta_0}$ (Amari, 2016). Introducing the $d \times m$ matrix $A_{\rho,\theta_0} = A_{\theta_0} D_{\rho,\theta_0}$ containing the scaled gradients $\rho(s)\nabla \pi_{\theta_0}(a|s)$ in its columns, we arrive at a local quadratic approximation of the objective

$$J(\theta; r^*) \approx \mathbb{E}_{\rho,\pi_{\theta_0}}[r^*(s,a)] + (\theta-\theta_0)^\top A_{\rho,\theta_0} r^* - \frac{\beta}{2}(\theta-\theta_0)^\top F_{\rho,\theta_0}(\theta-\theta_0).$$

**Remark 5.** *Just as described in Remark 2, the error in this local quadratic approximation of $J(\theta; r^*)$ can be controlled to a desired level of accuracy by taking $\beta$ to be sufficiently large; see Proposition 8.*

Based on this local quadratic approximation of the RLHF objective function, we can deduce (via first-order optimality conditions) a relation that suitably generalizes (2), between the optimal policy $\pi_{\theta^*}$ and the latent reward $r^*$, to parametric policy classes:

$$A_{\rho,\theta_0} r^* = \beta F_{\rho,\theta_0}(\theta^* - \theta_0) \iff \theta^* = \theta_0 + \frac{1}{\beta} F^{\dagger}_{\rho,\theta_0} A_{\rho,\theta_0} r^* . \tag{5}$$

This has the form of a natural policy gradient update (Kakade, 2001). Moreover, it partitions the set of all reward functions into equivalence classes as follows. For each policy parameter $\theta$, define

$$\mathcal{R}^{\beta}_{\text{eq}}(\theta) = \{r \in \mathbb{R}^m : A_{\rho,\theta_0} r = \beta F_{\rho,\theta_0}(\theta - \theta_0)\} . \tag{6}$$

**Lemma 6** (Equivalence classes induced by RLHF). *For a fixed $\theta \in \mathbb{R}^d$, two reward vectors $r_1, r_2 \in \mathbb{R}^m$ belong to $\mathcal{R}^{\beta}_{eq}(\theta)$ if and only if they differ by a vector $\delta \in \mathcal{N}(A_{\rho,\theta_0})$, i.e., a null space element of $A_{\rho,\theta_0}$.*

Note that for tabular policies, the class $\mathcal{R}^{\beta}_{\text{eq}}(\theta)$ *essentially* reduces to the singleton $r^{\beta}_{\theta}$, the "implicit reward" of DPO, i.e., $r^{\beta}_{\theta}(s, a) = \beta \log \frac{\pi_{\theta}(a|s)}{\pi_{\theta_0}(a|s)}$.[1]

Interestingly, we can show that DPO's linearized reward functions $\overline{r}^{\beta}_{\theta}$ are in bijective correspondence with RLHF's equivalence classes, with each linearized reward function being the minimum-norm representative of its equivalence class:

**Proposition 7** (Relationship between RLHF equivalence classes and DPO linearization). *For a base policy $\pi_{\theta_0}$ and KL penalty $\beta > 0$, the reward vector $r \in \mathcal{R}^{\beta}_{eq}(\theta)$ which has the minimum Mahalonobis-norm $\|r\|_{D_{\rho,\theta_0}} := \sqrt{r^\top D_{\rho,\theta_0} r}$, i.e., $\operatorname{argmin}_{r \in \mathbb{R}^m} \|r\|_{D_{\rho,\theta_0}}$ such that $A_{\rho,\theta_0} r = \beta F_{\rho,\theta_0}(\theta - \theta_0)$, is given by $\overline{r}^{\beta}_{\theta} = \beta A^\top_{\theta_0}(\theta - \theta_0)$, the local linearization of $r^{\beta}_{\theta}(s, a) = \beta \log \frac{\pi_{\theta}(a|s)}{\pi_{\theta_0}(a|s)}$.*

The following result helps to justify the local linear approximations made to (i) the DPO implicit reward function manifold and (ii) the RLHF policy optimization objective, by showing that error between the local approximations and the original functions can be controlled to an arbitrarily prescribed level by taking the deviation parameter $\beta$ to be suitably large (so that DPO and RLHF essentially reduce to optimization over policies in a neighborhood of $\pi_{\theta_0}$).

**Proposition 8** (Approximation errors). *Fix $\varepsilon > 0$. There exists a bounded neighborhood $\mathcal{E} \subset \mathbb{R}^d$ containing $\theta_0$, a bounded set $\mathcal{R} \subset \mathbb{R}^m$, and $\beta_{\min} > 0$ such that for every deviation parameter $\beta > \beta_{\min}$, we have (i) $r^{\beta}_{\theta} \in \mathcal{R}$ for each $\theta \in \mathcal{E}$, (ii) $r^{\beta}_{\text{DPO}} = r^{\beta}_{\theta}$ for some $\theta \in \mathcal{E}$, (iii) $r^{\beta}_{\theta}(s, a) - \beta \nabla \log \pi_{\theta_0}(a|s)^\top (\theta - \theta_0) \le \varepsilon$ for each $\theta \in \mathcal{E}$, and (iv) $J(\theta; r^*) - \left( \mathbb{E}_{\rho,\pi_{\theta_0}}[r^*(s, a)] + (\theta - \theta_0)^\top A_{\rho,\theta_0} r^* - \frac{\beta}{2}(\theta - \theta_0)^\top F_{\rho,\theta_0}(\theta - \theta_0) \right) \le \varepsilon$ for each $\theta \in \mathcal{E}$.*

### 4.2 THE AUXDPO ALGORITHM

We introduce a new direct alignment algorithm, AuxDPO, which leverages our insights from the analysis of the local geometry of DPO and RLHF policy optimization to mitigate the failure modes of DPO in a principled manner.

Recall that, in the general setting where the true reward function $r^*$ is misspecified (outside the implicit reward manifold $\mathcal{R}^{\beta}$), then DPO finds the optimal policy $\theta^*$ only if the reverse-KL projection of $r^*$ on $\mathcal{R}^{\beta}$ fortuitously lands on $r^{\beta}_{\theta^*}$ (Proposition 1). This depends crucially on relative proportions of pairwise preference counts $\{n_{i,j}\}_{i,j}$ in the dataset and, as such, is beyond the learner's control.

Instead, we take the following approach to encourage the optimization to move towards $r^{\beta}_{\theta^*}$. Note that by our local analysis of the RLHF optimization step (Sec 4.1), the true (misspecified) reward function $r^*$ and $r^{\beta}_{\theta^*}$ differ by an element of the nullspace of $A_{\rho,\theta_0}$, i.e., $r^* = r^{\beta}_{\theta^*} + \delta$, where

---

[1]More formally, $\mathcal{R}^{\beta}_{\text{eq}}(\theta)$ contains functions of the form $r^{\beta}_{\theta}(s, a) + \beta \log Z_{\theta}(s)$.

$\delta \in \mathcal{N}(A_{\rho,\theta_0}) \subsetneq \mathbb{R}^m$. Therefore, if we allow the search in the reward space $\mathbb{R}^m$ to utilize additional degrees of freedom $\delta$ along this nullspace (in addition to the usual degrees of freedom via $\theta$), then $r^*$ is no longer misspecified in this augmented representation. This should ideally result in the variables $\theta \in \mathbb{R}^d$ and $\delta \in \mathcal{N}(A_{\rho,\theta_0})$ settling in a manner that achieves $r^* = r^\beta_{\theta^*} + \delta^*$.

Observe that since $A_{\rho,\theta_0} = A_{\theta_0} D_{\rho,\theta_0}$, both $\mathcal{N}(A_{\theta_0})$ and $\mathcal{N}(A_{\rho,\theta_0})$ have the same dimension. Moreover, $\mathcal{C}(A_{\theta_0}^\top)$ and $\mathcal{C}(A_{\theta_0})$ also have same dimension. Thus, by the rank-nullity theorem, by varying both $\theta$ and $\delta$, we can search over the entire space of the rewards $\mathbb{R}^m$, contrary to DPO which searches only over $\mathcal{C}(A_{\theta_0}^\top)$ (under linear approximation), a manifold in $\mathbb{R}^m$ with dimension at most $d$.

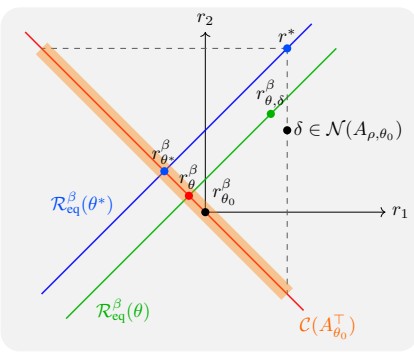

To this end, we introduce auxiliary variables $\delta \in \mathbb{R}^m$ into the population loss of DPO (3), and minimize it jointly over $\theta \in \mathbb{R}^d, \delta$ while enforcing the nullspace constraint $\delta \in \mathcal{N}(A_{\rho,\theta_0})$. This gives us the AuxDPO procedure[2]:

$$\underset{\theta \in \mathbb{R}^d, \delta \in \mathcal{N}(A_{\rho,\theta_0})}{\text{minimize}} \mathcal{L}(\theta, \delta), \quad \text{where}$$

$$\mathcal{L}(\theta, \delta) = -\sum_{s,a,a'} n_{s,a,a'} \Big[ p^{\text{BTL}}_{s,a,a'}(r^*) \log p^{\text{BTL}}_{s,a,a'}(r^\beta_{\theta,\delta}) \quad (7)$$
$$+ (1 - p^{\text{BTL}}_{s,a,a'}(r^*)) \log \left(1 - p^{\text{BTL}}_{s,a,a'}(r^\beta_{\theta,\delta})\right) \Big].$$

**Proposition 9** (Auxiliary variables bypass misspecification). *Let the hypothesis of Proposition 1 hold. Fix a tolerance $\varepsilon > 0$. Then, for sufficiently large $\beta > 0$, the optimization* (7) *is minimized at $\theta = \theta^*$ up to error $O(\varepsilon)$.*

We develop AuxDPO's corresponding empirical loss version, which can be implemented with a finite preference dataset. We convert the constrained optimization over the set $\mathcal{N}(A_{\rho,\theta_0}) \subset \mathbb{R}^m$ to an unconstrained one over $\mathbb{R}^m$ by

Figure 3: AuxDPO fixes DPO's misspecification. $r^*$ is the latent reward. The blue line denotes the equivalence class $\mathcal{R}^\beta_{\text{eq}}(\theta^*)$ of all reward functions that yield the RLHF-optimal policy $\pi_{\theta^*}$. The red line denotes the linear approximation $\mathcal{C}(A_{\theta_0}^\top)$ of the implicit reward manifold $\mathcal{R}^\beta$. The region shaded in orange represents all possible implicit reward functions that DPO can possibly project onto. The green line depicts the domain of optimization over AuxDPO's auxiliary variables $\delta \in \mathcal{N}(A_{\rho,\theta_0})$ for a fixed $\theta$ (the line shifts in parallel for other $\theta$). $\delta$ introduces additional degrees of freedom, which help push the KL projection of $r^*$ to lie in the equivalence class $\mathcal{R}_{\theta^*}$. The projection induces the optimal policy $\pi_{\theta^*}$.

adding the penalty term $\|A_{\rho,\theta_0} \delta\|_2^2$ to the log-loss. Note that $A_{\rho,\theta_0}\delta = \mathbb{E}_{\rho,\pi_{\theta_0}}[\delta(s,a)\nabla \log \pi_{\theta_0}(a|s)]$. We approximate it with a given dataset $\mathcal{D} = (s^{(i)}, a_w^{(i)}, a_l^{(i)})_{i=1}^n$ in Monte-Carlo fashion. This leads to the empirical AuxDPO loss $\mathcal{L}_\mathcal{D}$ over variables $\theta \in \mathbb{R}^d$ and $\delta \in \mathbb{R}^{2n}$:

$$\mathcal{L}_\mathcal{D}(\theta, \delta) = -\frac{1}{n} \sum_{i=1}^n \log \sigma \left( r^\beta_\theta(s^{(i)}, a_w^{(i)}) - r^\beta_\theta(s^{(i)}, a_l^{(i)}) + \delta(s^{(i)}, a_w^{(i)}) - \delta(s^{(i)}, a_l^{(i)}) \right)$$
$$+ \lambda \left\| \frac{1}{2n} \sum_{i=1}^n \left( \delta(s^{(i)}, a_w^{(i)}) \nabla \log \pi_{\theta_0}(a_w^{(i)} \mid s^{(i)}) + \delta(s^{(i)}, a_l^{(i)}) \nabla \log \pi_{\theta_0}(a_l^{(i)} \mid s^{(i)}) \right) \right\|^2.$$

where $\delta = \left\{ \delta(s^{(i)}, a_w^{(i)}), \delta(s^{(i)}, a_l^{(i)}) \right\}_{i=1}^n \in \mathbb{R}^{2n}$ denotes the vector of auxiliary variables (typically $2n \ll m$) and $\lambda > 0$ is a hyper-parameter that is responsible for enforcing the nullspace constraint on $\delta$. Note that the total number of trainable parameters is $d + 2n = O(d)$, since typically, $n \ll d$.

## 5 EXPERIMENTS

**Datasets.** We conduct evaluations on two benchmark datasets: REWARDBENCH V2 and MMLU-PRO. REWARDBENCH V2 (Malik et al., 2025) contains $1.87K$ prompts covering categories like factuality, precise instruction following, and focus, with each prompt containing a chosen and a rejected response. MMLU-PRO (Wang et al., 2024b) is a multi-task understanding dataset containing $12K$ complex questions across various disciplines. Each question has 10 possible answers and a

---

[2]AuxDPO can be run on any dataset, irrespective of any approximation needing to be assumed. Its theoretical guarantee (Prop. 9), however, is valid in the local sense, which manifests at sufficiently high $\beta$.

| Model | Dataset | Method | DPO | AuxDPO | IPO | DPOP |
|-------|---------|--------|-----|--------|-----|------|
| Llama3.1-8B | MMLU-PRO | ID | 57.14 | **63.26** | 59.18 | 61.22 |
| | MMLU-PRO | OOD | 8.16 | **14.28** | 10.20 | 6.12 |
| | REWARDBENCH V2 | ID | 56.01 | **66.72** | 61.34 | 62.27 |
| | REWARDBENCH V2 | OOD | 14.31 | **32.44** | 20.17 | 19.87 |
| Llama3.2-1B | MMLU-PRO | ID | 39.58 | **45.83** | 43.75 | 44.21 |
| | MMLU-PRO | OOD | 6.25 | 12.52 | **14.58** | 4.16 |
| | REWARDBENCH V2 | ID | 77.21 | **86.37** | 69.72 | 71.21 |
| | REWARDBENCH V2 | OOD | 14.11 | **43.27** | 20.42 | 18.76 |
| Qwen3-0.6B | MMLU-PRO | ID | 53.12 | **61.78** | 47.48 | 56.67 |
| | MMLU-PRO | OOD | 11.34 | **22.22** | 15.56 | 17.78 |
| | REWARDBENCH V2 | ID | 55.10 | **65.31** | 53.06 | 51.02 |
| | REWARDBENCH V2 | OOD | −8.16 | **18.36** | −8.23 | −6.25 |

Table 1: Algorithm comparison. Values show percentage change in mean accuracy relative to the base policy, across MMLU-PRO and REWARDBENCH V2 under in-domain (ID) and out-of-domain (OOD) settings. Best gains are in **bold**, second-best are underlined. Accuracies which degrade from the base policy are marked in red.

correct answer. We use ULTRAFEEDBACK (Cui et al., 2024) as our training dataset. Specifically, its pre-processed and binarized version (Dong et al., 2024), which generates higher-quality reward models (Wang et al., 2024a; Xiong et al., 2024; Banerjee and Gopalan, 2024).

| **Subject** | **Base** | **DPO** | | **AuxDPO** | | **IPO** | | **DPOP** | |
|-------------|----------|---------|-----|------------|-----|---------|-----|----------|-----|
| | | OOD | ID | OOD | ID | OOD | ID | OOD | ID |
| Overall | 25.37 | 27.06 | 46.60 | **39.26** | **51.95** | 31.73 | 47.17 | 32.64 | 48.25 |
| Biology | 55.93 | 52.44 | 76.50 | **75.07** | **86.87** | 60.67 | 79.12 | 60.39 | 81.50 |
| Business | 13.18 | 20.03 | 35.67 | **29.96** | **43.49** | 24.21 | 32.56 | 24.08 | 37.96 |
| Chemistry | 10.42 | 16.25 | 37.12 | **26.78** | **40.86** | 21.64 | 34.56 | 21.73 | 35.86 |
| Comp. Sc. | 24.39 | 26.59 | 42.78 | **38.93** | **48.35** | 31.46 | 43.57 | 31.22 | 47.08 |
| Economics | 38.98 | 37.80 | 59.24 | **55.12** | **68.30** | 44.55 | 62.58 | 43.60 | 64.50 |
| Engineering | 10.22 | 17.75 | 46.06 | **32.18** | **48.60** | 26.01 | 41.46 | 28.48 | 43.82 |
| Health | 41.81 | 41.32 | 62.41 | **55.67** | **68.58** | 44.99 | 64.08 | 44.13 | 66.23 |
| History | 38.06 | 32.55 | 54.96 | **47.41** | **64.58** | 38.32 | 59.56 | 40.42 | 57.29 |
| Law | 27.25 | 23.61 | 42.20 | **34.50** | **48.01** | 27.88 | 45.25 | 30.43 | 42.31 |
| Math | 11.77 | 16.51 | 28.97 | **22.26** | **30.80** | 17.99 | 29.34 | 20.28 | 30.24 |

Table 2: Per-subject accuracies (top 10 subjects alphabetically) and overall win-rates across baseline (Llama3.1-8B) and preference optimization methods. For each method, two settings are shown: **OOD** (cross-domain transfer) and **ID** (in-domain learning), along with the reported results. In each row, the best accuracy is shown in **bold**, and the second-best is underlined. Accuracies which degrade from the base policy are marked in red.

**Evaluation and Methodology.** We compare AuxDPO with DPO, IPO (Azar et al., 2023), and DPOP (Pal et al., 2024). Table 1 reports accuracies in terms of whether the logits of the chosen response were higher than those of the rejected response. While REWARDBENCH V2 provides chosen and rejected responses directly, we make MMLU-PRO into a preference dataset by filtering the correct answer as the chosen response and any incorrect response as the rejected response. We consider both in-distribution (ID) and out-of-distribution (OOD) evaluation settings. In the ID setup, each dataset is split 80/20 into training and evaluation subsets, ensuring IID comparisons. In the OOD setup, models are trained on cleaned ULTRAFEEDBACK and evaluated on the preference datasets. We report full finetuning results on the models. Table 2 presents overall and subject-wise accuracies on MMLU-PRO. Accuracy is measured by comparing the finetuned model's generated answer with the correct answer provided in the dataset. We compare all methods on Llama3.1-8B under both OOD and ID settings. Across all three models, we see that AuxDPO outperforms other finetuning methods. Ablation studies, implementation, and dataset details are presented in Appendix B.2.

ACKNOWLEDGMENTS

SRC would like to thank an early-career research grant from ANRF, India. DB would like to thank HP AI Labs for providing the necessary computational infrastructure to conduct the experiments.

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
