## A   MISSING PROOFS

In this section, we restate our theoretical results and provide their proofs.

**Proposition 1** (DPO is weighted KL-projection). *Assume that the pairwise preference data are drawn from $p_{s,a_w,a_l}^{\text{BTL}}(r^*)$ for some $r^* \in \mathbb{R}^m$, with $n_{s,a,a'}$ preference pairs drawn for each triplet $(s,a,a')$. If $\theta_{DPO}$ minimizes the DPO loss (3), then its corresponding implicit reward function satisfies*

$$r_{\theta_{DPO}}^{\beta} = \arg \min_{r \in \mathcal{R}^{\beta}} \sum_{s,a,a'} n_{s,a,a'} \, d_{KL}\left(p_{s,a,a'}^{\text{BTL}}(r^*) \| p_{s,a,a'}^{\text{BTL}}(r)\right) , \tag{4}$$

*where $d_{KL}(p\|q)$ is the Kullback-Leibler divergence between two Bernoulli random variables with parameters $p, q$.*

*Proof.* First note that minimizing $\mathcal{L}(\theta)$ is equivalent to minimizing the expected negative log-likelihood, i.e.,

$$\min_{\theta \in \mathbb{R}^d} - \sum_{s,a,a'} n_{s,a,a'} \left[ p_{s,a,a'}^{\text{BTL}}(r^*) \log p_{s,a,a'}^{\text{BTL}}(r_\theta^\beta) + (1 - p_{s,a,a'}^{\text{BTL}}(r^*)) \log \left(1 - p_{s,a,a'}^{\text{BTL}}(r_\theta^\beta)\right) \right] .$$

Note that

$$p_{s,a,a'}^{\text{BTL}}(r^*) \log p_{s,a,a'}^{\text{BTL}}(r_\theta^\beta) + (1 - p_{s,a,a'}^{\text{BTL}}(r^*)) \log \left(1 - p_{s,a,a'}^{\text{BTL}}(r_\theta^\beta)\right)$$
$$= -d_{\text{KL}}\left(p_{s,a,a'}^{\text{BTL}}(r^*) \| p_{s,a,a'}^{\text{BTL}}(r_\theta^\beta)\right) + H(p_{s,a,a'}^{\text{BTL}}(r^*)),$$

where $d_{\text{KL}}(p\|q)$ denotes the KL divergence between two Bernoulli random variables with parameters $p, q$, and $H(p)$ denotes the entropy of a Bernoulli random variable with parameter $p$. Since $H(p_{s,a,a'}^{\text{BTL}}(r^*))$ is constant with respect to $\theta$, the loss minimization is equivalent to minimizing the reverse KL divergence:

$$\min_{\theta \in \mathbb{R}^d} \sum_{s,a,a'} n_{s,a,a'} \cdot d_{\text{KL}}\left(p_{s,a,a'}^{\text{BTL}}(r^*) \| p_{s,a,a'}^{\text{BTL}}(r_\theta^\beta)\right) .$$

Since $\mathcal{R}^\beta = \{r_\theta^\beta : \theta \in \mathbb{R}^d\}$, this is equivalent to solving

$$\min_{r \in \mathcal{R}^\beta} \sum_{s,a,a'} n_{s,a,a'} \cdot d_{\text{KL}}\left(p_{s,a,a'}^{\text{BTL}}(r^*) \| p_{s,a,a'}^{\text{BTL}}(r)\right) ,$$

which completes the proof. $\qquad\square$

**Lemma 6** (Equivalence classes induced by RLHF). *For a fixed $\theta \in \mathbb{R}^d$, two reward vectors $r_1, r_2 \in \mathbb{R}^m$ belong to $\mathcal{R}_{eq}^\beta(\theta)$ if and only if they differ by a vector $\delta \in \mathcal{N}(A_{\rho,\theta_0})$, i.e., a null space element of $A_{\rho,\theta_0}$.*

*Proof.* The result holds since

$$A_{\rho,\theta_0} r_1 = A_{\rho,\theta_0} r_2 \iff A_{\rho,\theta_0}(r_1 - r_2) = 0 \iff r_1 - r_2 = \delta \in \mathcal{N}(A_{\rho,\theta_0}).$$

which completes the proof. $\qquad\square$

**Proposition 7** (Relationship between RLHF equivalence classes and DPO linearization). *For a base policy $\pi_{\theta_0}$ and KL penalty $\beta > 0$, the reward vector $r \in \mathcal{R}_{eq}^\beta(\theta)$ which has the minimum Mahalonobis-norm $\|r\|_{D_{\rho,\theta_0}} := \sqrt{r^\top D_{\rho,\theta_0} r}$, i.e., $\operatorname{argmin}_{r \in \mathbb{R}^m} \|r\|_{D_{\rho,\theta_0}}$ such that $A_{\rho,\theta_0} r = \beta F_{\rho,\theta_0}(\theta - \theta_0)$, is given by $\overline{r}_\theta^\beta = \beta A_{\theta_0}^\top(\theta - \theta_0)$, the local linearization of $r_\theta^\beta(s,a) = \beta \log \frac{\pi_\theta(a|s)}{\pi_{\theta_0}(a|s)}$.*

*Proof.* To this end, a representative $\overline{r}_\theta^\beta$ can be the reward vector $r \in \mathcal{R}_{eq}^\beta(\theta)$ which has the minimum Mahalonobis-norm $\|r\|_{D_{\rho,\theta_0}}$, i.e.,

$$\overline{r}_\theta^\beta = \operatorname*{argmin}_{r \in \mathbb{R}^m} \|r\|_{D_{\rho,\theta_0}} \quad \text{s.t.} \quad A_{\rho,\theta_0} r = \beta F_{\rho,\theta_0}(\theta - \theta_0) , \tag{8}$$

where $D_{\rho,\theta_0}$ is an $m \times m$ diagonal matrix ($m = |\mathcal{S}| \cdot |\mathcal{A}|$), whose diagonal entries are indexed by $\{\rho(s)\pi_{\theta_0}(a|s)\}_{s,a}$.

Using the standard Lagrange multiplier technique, one can solve (8) to get

$$\bar{r}_\theta^\beta = \beta D_{\rho,\theta_0}^{-1} A_{\rho,\theta_0}^\top (A_{\rho,\theta_0} D_{\rho,\theta_0}^{-1} A_{\rho,\theta_0}^\top)^{-1} F_{\rho,\theta_0}(\theta - \theta_0) = \beta D_{\rho,\theta_0}^{-1} A_{\rho,\theta_0}^\top (\theta - \theta_0) \,, \tag{9}$$

where the last equality is because the Fisher information matrix satisfies

$$F_{\rho,\theta} = \mathbb{E}_{s\sim\rho(\cdot),a\sim\pi_\theta(\cdot|s)} \left[ \nabla \log \pi_\theta(a|s) \nabla \log \pi_\theta(a|s)^\top \right]$$

$$= \sum_{s,a} \frac{\rho(s)}{\pi_\theta(a|s)} \nabla \pi_\theta(a|s) \nabla \pi_\theta(a|s)^\top = A_{\rho,\theta} D_{\rho,\theta}^{-1} A_{\rho,\theta}^\top \,.$$

This completes the proof. $\qquad\square$

**Proposition 8** (Approximation errors). *Fix $\varepsilon > 0$. There exists a bounded neighborhood $\mathcal{E} \subset \mathbb{R}^d$ containing $\theta_0$, a bounded set $\mathcal{R} \subset \mathbb{R}^m$, and $\beta_{\min} > 0$ such that for every deviation parameter $\beta > \beta_{\min}$, we have (i) $r_\theta^\beta \in \mathcal{R}$ for each $\theta \in \mathcal{E}$, (ii) $r_{\mathrm{DPO}}^\beta = r_\theta^\beta$ for some $\theta \in \mathcal{E}$, (iii) $r_\theta^\beta(s,a) - \beta\nabla\log\pi_{\theta_0}(a|s)^\top(\theta - \theta_0) \leq \varepsilon$ for each $\theta \in \mathcal{E}$, and (iv) $J(\theta; r^*) - \left( \mathbb{E}_{\rho,\theta_0}[r^*(s,a)] + (\theta - \theta_0)^\top A_{\rho,\theta_0} r^* - \frac{\beta}{2}(\theta - \theta_0)^\top F_{\rho,\theta_0}(\theta - \theta_0) \right) \leq \varepsilon$ for each $\theta \in \mathcal{E}$.*

*Proof.* Define $f : \mathbb{R}^m \to \mathbb{R}$ by $f(r) := \sum_{s,a,a'} n_{s,a,a'} d_{\mathrm{KL}} \left( p_{s,a,a'}^{\mathrm{BTL}}(r^*) \| p_{s,a,a'}^{\mathrm{BTL}}(r) \right)$. Let $\mathcal{R} := \cup_{\beta>0} \{ r \in \mathcal{R}^\beta : f(r) \leqslant f(0) \}$. It follows from (4) that for each $\beta$, $r_{\mathrm{DPO}}^\beta \in \mathcal{R}$, establishing **Part (ii)**. Furthermore, $\mathcal{R}$ is a bounded set since each of the sets in the union is bounded. Let $\gamma$ be a bound on the $\ell_2$-norm of any reward vector in $\mathcal{R}$ (note that $\gamma$ is independent of any deviation parameter $\beta > 0$).

**Part (i):** The function $\theta \mapsto r_\theta^\beta$ is assumed to be smooth. Also, $r_{\theta_0}^\beta = 0$. Since $\mathcal{R}$ is bounded, there exists a neighborhood $\mathcal{E}$ of $\theta_0$ such that $r_\theta^\beta \in \mathcal{R}$ for all $\theta \in \mathcal{E}$.

**Part (iii):** For $\theta \in \mathcal{E}$, using first-order Taylor series approximation of $r_\theta^\beta(s,a)$ around $\theta_0$, we get the approximation error

$$e_r(\theta) := r_\theta^\beta(s,a) - \beta\nabla\log\pi_{\theta_0}(a|s)^\top(\theta - \theta_0) = \frac{\beta}{2}(\theta - \theta_0)^\top \nabla^2 \log \pi_{\bar\theta}(a|s)(\theta - \theta_0)$$

for some $\bar\theta$ in the line segment joining $\theta$ and $\theta_0$.

Suppose the log-likelihood function $\log \pi_\theta(a|s)$ is $L$-smooth in $\theta$ for every $(s,a)$, i.e.

$$\|\nabla^2 \log \pi_\theta(a|s)\|_{\mathrm{op}} \leq L \quad \text{for all } (s,a), \theta,$$

where $\|\cdot\|_{\mathrm{op}}$ denotes the spectral (operator) norm. Then every eigenvalue of the Hessian $H_\theta(s,a) := \nabla^2 \log \pi_\theta(s,a)$ lies in the interval $[-L, L]$. In particular, $\lambda_{\max}(H_\theta(s,a)) \leq L$. Using this, the error can be upper-bounded as

$$e_r(\theta) \leq \frac{\beta L}{2} \|\theta - \theta_0\|^2$$

Let $s_\theta(s,a) = \nabla_\theta \log \pi_\theta(a|s)$ be the score function. Assume $\|s_\theta(s,a)\| \leq G$. Then we have

$$r_\theta^\beta(s,a) \leq \beta G \|\theta - \theta_0\| + \frac{\beta L}{2} \|\theta - \theta_0\|^2 = O(\beta \|\theta - \theta_0\|) \,.$$

Furthermore, since $\left\| r_\theta^\beta \right\| \leq \gamma = O(1)$ for $\theta \in \mathcal{E}$, it ensures that $\|\theta - \theta_0\| = O(1/\beta)$. This further yields the approximation error

$$e_r(\theta) \leq O(1/\beta) \leq \varepsilon \,,$$

for $\beta > \beta_{\min}$ (which depends on $\varepsilon$ and constants $L, G$).

**Part (iv):** For $\theta \in \mathcal{E}$, using first order Taylor series approximation of $r_\theta^\beta(s,a)$ around $\theta_0$, we get the approximation error for the expected reward

$$e_{\rho,r}(\theta) := \mathbb{E}_{\rho,\pi_\theta}[r^*(s,a)] - r^{*\top}\pi_{\theta_0} + (\theta - \theta_0)^\top A_{\rho,\theta_0} r^* = \frac{1}{2}(\theta - \theta_0)^\top H_{\rho,\bar\theta}^{(r)}(\theta - \theta_0) \,,$$

for some $\bar{\theta}$ between $\theta_0$ and $\theta$. Here

$$H_{\rho,\bar{\theta}}^{(r)} = \sum_{s,a} \rho(s) r^*(s,a) \pi_\theta(a|s) \big[ s_\theta(s,a) s_\theta(s,a)^\top + H_\theta(s,a) \big],$$

where $s_\theta(s,a) = \nabla_\theta \log \pi_\theta(a|s)$ and $H_\theta(s,a) = \nabla_\theta^2 \log \pi_\theta(a|s)$. Assume $\|s_\theta(s,a)\| \leq G$, $\|H_\theta(s,a)\|_{\mathrm{op}} \leq L$, $|r^*(s,a)| \leq R_{\max}$. Then $\|H_{\rho,\bar{\theta}}^{(r)}\|_{\mathrm{op}} \leq R_{\max}(G^2 + L) := M_1$ yielding

$$|e_{\rho,r}(\theta)| \leq \frac{M_1}{2} \|\theta - \theta_0\|^2.$$

Similarly, using second order Taylor series approximation of $\mathbb{E}_\rho\big[D_{\mathrm{KL}}(\pi_\theta(\cdot|s)\|\pi_{\theta_0}(\cdot|s))\big]$ around $\theta_0$, we get the approximation error

$$e_{\rho,\mathrm{kl}}(\theta) := \mathbb{E}_\rho\big[D_{\mathrm{KL}}(\pi_\theta(\cdot|s)\|\pi_{\theta_0}(\cdot|s))\big] - \frac{1}{2}(\theta - \theta_0)^\top F_{\rho,\theta_0}(\theta - \theta_0)$$

$$= \frac{1}{6} H_{\rho,\tilde{\theta}}^{(kl)}[(\theta - \theta_0), (\theta - \theta_0), (\theta - \theta_0)]$$

for some $\tilde{\theta}$ between $\theta_0$ and $\theta$. Assume $\|T_\theta(s,a)\|_{\mathrm{op}} \leq Q$ for $T_\theta(s,a) = \nabla_\theta^3 \log \pi_\theta(a|s)$, then $\|H_{\rho,\theta}^{(kl)}\|_{\mathrm{op}} \leq G^3 + C_1 L G + C_2 Q =: M_2$, with modest constants $C_1, C_2$, yielding

$$|e_{\rho,\mathrm{kl}}(\theta)| \leq \frac{M_2}{6} \|\theta - \theta_0\|^3.$$

Therefore, we get the total error in approximating the objective $J(\theta; r^*)$:

$$\left| J(\theta; r^*) - \left( \mathbb{E}_{\rho,\pi_{\theta_0}}[r^*(s,a)] + (\theta - \theta_0)^\top A_{\rho,\theta_0} r^* - \frac{\beta}{2}(\theta - \theta_0)^\top F_{\rho,\theta_0}(\theta - \theta_0) \right) \right|$$

$$\leq \frac{M_1}{2} \|\theta - \theta_0\|^2 + \frac{\beta M_2}{6} \|\theta - \theta_0\|^3 = O\left( \|\theta - \theta_0\|^2 + \beta \|\theta - \theta_0\|^3 \right) = O(1/\beta^2)$$

since $\|\theta - \theta_0\| = O(1/\beta)$ for $\theta \in \mathcal{E}$. Thus the error $\leq \varepsilon$ for some $\beta > \beta_{\min}$. $\qquad \square$

**Proposition 9** (Auxiliary variables bypass misspecification). *Let the hypothesis of Proposition 1 hold. Fix a tolerance $\varepsilon > 0$. Then, for sufficiently large $\beta > 0$, the optimization (7) is minimized at $\theta = \theta^*$ up to error $O(\varepsilon)$.*

*Proof.* First note that minimizing the objective in (7) is equivalent to minimizing the reverse KL divergence, i.e.,

$$\min_{\theta \in \mathbb{R}^d, \delta \in \mathcal{N}(A_{\rho,\theta_0})} - \sum_{s,a,a'} n_{s,a,a'} \left[ p_{s,a,a'}^{\mathrm{BTL}}(r^*) \log p_{s,a,a'}^{\mathrm{BTL}}(r_{\theta,\delta}^\beta) + (1 - p_{s,a,a'}^{\mathrm{BTL}}(r^*)) \log \left( 1 - p_{s,a,a'}^{\mathrm{BTL}}(r_{\theta,\delta}^\beta) \right) \right]$$

$$= \min_{\theta \in \mathbb{R}^d, \delta \in \mathcal{N}(A_{\rho,\theta_0})} \sum_{s,a,a'} n_{s,a,a'} \cdot d_{\mathrm{KL}} \left( p_{s,a,a'}^{\mathrm{BTL}}(r^*) \| p_{s,a,a'}^{\mathrm{BTL}}(r_{\theta,\delta}^\beta) \right),$$

where $r_{\theta,\delta}^\beta(s,a) = r_\theta^\beta(s,a) + \delta(s,a)$. Since $\{r_{\theta,\delta}^\beta : \theta \in \mathbb{R}^d, \delta \in \mathcal{N}(A_{\rho,\theta_0})\} = \mathbb{R}^m$, this is equivalent to solving

$$\min_{r \in \mathbb{R}^m} \sum_{s,a,a'} n_{s,a,a'} \cdot d_{\mathrm{KL}} \left( p_{s,a,a'}^{\mathrm{BTL}}(r^*) \| p_{s,a,a'}^{\mathrm{BTL}}(r) \right).$$

Because KL divergence is nonnegative and equals zero iff its arguments coincide, the above objective is minimized (to zero) if and only if $p_{s,a,a'}^{\mathrm{BTL}}(r) = p_{s,a,a'}^{\mathrm{BTL}}(r^*)$, $\forall s, \forall a \neq a'$.

From Proposition 8, since $r_\theta^\beta(s,a) - \bar{r}_\theta^\beta(s,a) \leq \varepsilon$ for each $\theta \in \mathcal{E}$, where $\bar{r}_\theta^\beta \in \mathcal{C}(A_{\theta_0}^\top)$, it holds that the minimizer $r = r_{\theta^*,\delta^*}^\beta$ upto an order $O(\varepsilon)$. $\qquad \square$

# B ADDITIONAL EXPERIMENTAL DETAILS

## B.1 ABLATION STUDY ON TRAINABLE PARAMETERS

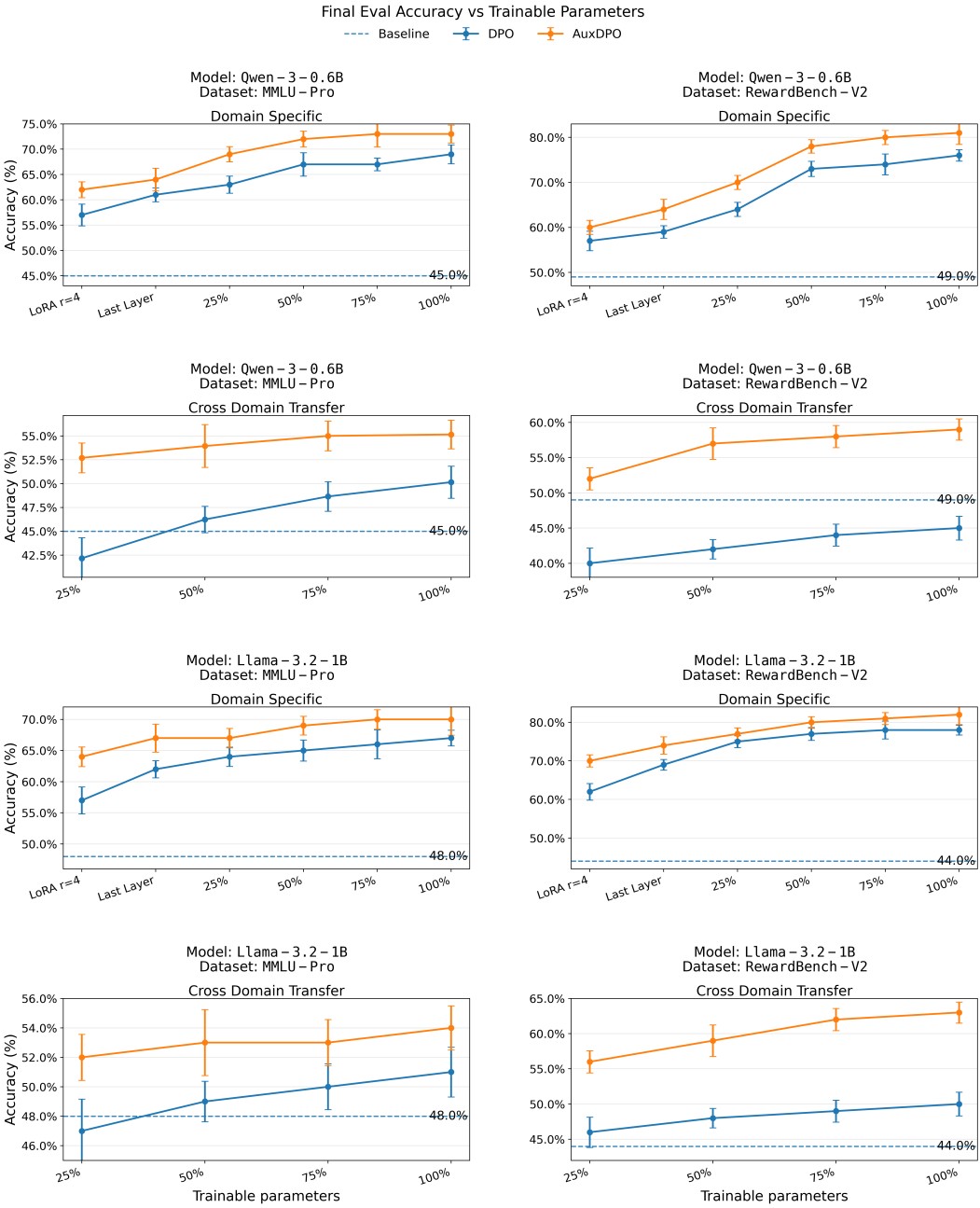

Figure 4: Final evaluation accuracy (%) for `Qwen-3-0.6B` and `Llama-2.3-1B` on `MMLU-Pro` and `RewardBench-V2` under *Domain-Specific* (ID) and *Cross-Domain Transfer* (OOD) settings. Each subplot compares DPO and AuxDPO across fractions of trainable parameters (25%, 50%, 75%, 100%); for ID panels we also include `LoRA r=4` and `Last Layer` configurations. Markers show run means with error bars denoting ±1 std, and dashed lines (when present) indicate per-panel baselines.

In this section, we perform ablation studies on the performance of AusxDPO vs DPO by changing the number of trainable parameters. Given that AuxDPO mitigates model misspecification issues that can

arise while using standard DPO, we focus mostly on having a low number of trainable parameters, where model expressibility is generally lower than entire fine-tuned models.

**Datasets** We conduct evaluations on two benchmark datasets: REWARDBENCH V2 and MMLU-PRO. REWARDBENCH V2 (Malik et al., 2025) is a multi-skill reward modeling benchmark designed to bring new, challenging data for accuracy-based reward model evaluation. The dataset contains around $1.87K$ prompts covering categories as factuality, precise instruction following, focus with each prompt containing a chosen and an accepted response. MMLU-PRO (Wang et al., 2024b) is a robust and challenging massive multi-task understanding dataset tailored to more rigorously benchmark large language models' capabilities. This dataset contains $12K$ complex questions across various disciplines. It's dataset contains around $12K$ questions ranging across fields such as math, law, chemistry, business, history, and psychology. For each question, there is a list of 10 possible correct answers and the corresponding correct answer. We use ULTRAFEEDBACK (Cui et al., 2024) as our training dataset. Specifically, we use the pre-processed and binarized version of ULTRAFEEDBACK as presented by Dong et al. (2024), which has been shown to generate higher quality reward models (Dong et al., 2024; Wang et al., 2024a; Xiong et al., 2024; Banerjee and Gopalan, 2024). REWARDBENCH V2 is a preference dataset and provides chosen and rejected responses and is used as. MMLU-PRO by default is not a preference dataset. We make it into a preference dataset by filtering the correct answer as the chosen response and any incorrect response as the rejected response.

**Methodology** We consider both in-distribution (ID) and out-of-distribution (OOD) evaluation settings. In the ID setup, each dataset is split 80/20 into training and evaluation subsets, ensuring IID comparisons. In the OOD setup, models are trained on the cleaned ULTRAFEEDBACK dataset and evaluated on the held-out preference datasets. We vary the fraction of trainable parameters across $25\%, 50\%, 75\%, 100\%$, by unfreezing the last $k\%$ of transformer blocks (by depth). In addition, for the in-domain (ID) panels we include two constrained-capacity baselines: `Last Layer`, where only the last transformer block is trainable and `LoRA r=4`, low-rank adapters inserted in the last transformer block on the `q_proj, k_proj, v_proj, o_proj` matrices. Unless otherwise noted, optimization settings, token budgets, and data splits are held fixed within each model–dataset panel so that the *only* differences are (i) the algorithm (DPO, AuxDPO, IPO, DPOP) and (ii) the trainable-parameter configuration. Each marker reports the mean across 20 random seeds and error bars denote $\pm 1$ standard deviation; dashed lines indicate the per-panel base policy.

**Evaluation.** As we decrease the number of trainable parameters, we perform finetuning by either ID or OOD method and evaluate accuracies by measuring the log probabilities of the chosen and rejected responses. Figure 4 reports final evaluation accuracy (%) for `Qwen-3-0.6B` and `Llama-2.3-1B` across MMLU-PRO and REWARDBENCH-V2 under ID and OOD. Each subplot compares DPO vs. AuxDPO at $25, 50, 75, 100\%$ trainable parameters; ID panels additionally include `LoRA r=4` and `Last Layer`. Figure 5 reports the analogous sweep for `Llama-2.1-8B`, now comparing DPO, AuxDPO, IPO, and DPOP at the same trainable fractions (ID/OOD), with dashed baselines and $\pm 1$ std error bars. Together, the figures isolate (a) sensitivity to optimization method, (b) sensitivity to effective capacity, and (c) ID vs. OOD robustness, while controlling for data and compute.

**Results.** Within a panel, horizontal movement (left→right) reflects increasing capacity (more parameters unfrozen); vertical separation between method curves reflects algorithmic gains at fixed capacity. The dashed line is the per-panel base policy; curves above (below) it indicate improvement (degradation). In ID panels, compare `LoRA r=4` and `Last Layer` against the fractional unfreeze settings to understand cost–performance trade-offs at very low trainable budgets.

**Empirical trends (qualitative).** Across models and datasets we generally observe: (i) accuracy tends to improve as the trainable fraction increases, with diminishing returns beyond $75\%$; (ii) AuxDPO often outperforms vanilla DPO at matched capacity, particularly on REWARDBENCH-V2; (iii) AuxDPO gains for OOD are more significant than than ID gains, reflecting the difficulty of cross-domain generalization for standard DPO; and (iv) At lower capacity (`LoRA r=4` and `Last Layer`) AuxDPO performs significantly better than DPO.

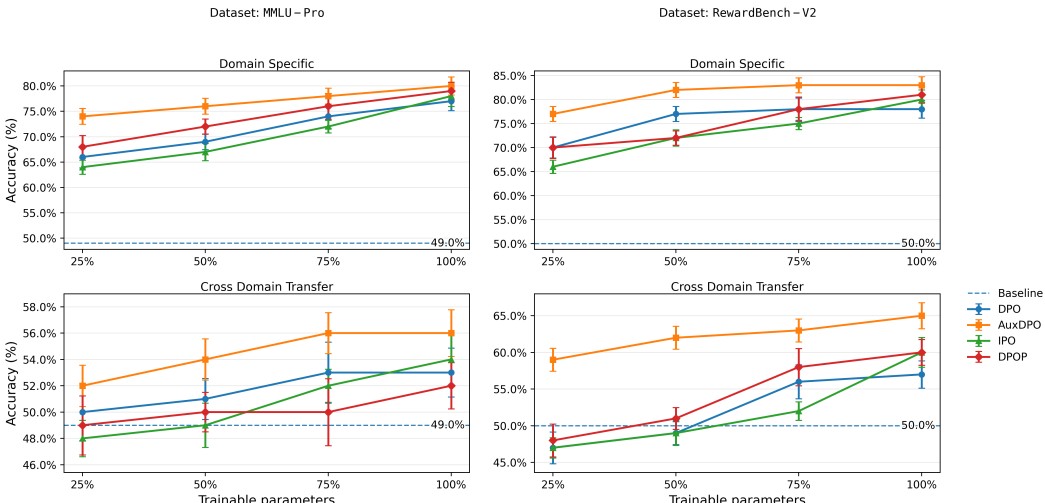

Figure 5: Final evaluation accuracy (%) of the `Llama-2.1-8B` model on `MMLU-Pro` and `RewardBench-V2` under *Domain-Specific* (ID) and *Cross-Domain Transfer* (OOD) settings. Each subplot compares DPO, AuxDPO, IPO, and DPOP across fractions of trainable parameters (25%, 50%, 75%, 100%); markers show run means with error bars denoting $\pm 1$ std, and a dashed line marks the per-panel baseline.

## B.2 IMPLEMENTATION DETAILS OF AUXDPO

**Objects and notation.** Given a preference dataset $\big(s^{(i)}, a_w^{(i)}, a_l^{(i)}\big)_{i=1}^n$, AuxDPO introduces a per-example offset vector $\delta \in \mathbb{R}^{2n}$ with

$$\delta_{2i-1} = \delta\big(s^{(i)}, a_w^{(i)}\big), \qquad \delta_{2i} = \delta\big(s^{(i)}, a_l^{(i)}\big) \quad (i = 1, \ldots, n).$$

Let $\theta_0$ denote the reference policy and let $d$ be the number of *trainable* parameters. We define $A_{\theta_0} \in \mathbb{R}^{d \times 2n}$ whose columns are reference-model gradients on *response tokens only*:

$$A_{\theta_0}[:, 2i-1] = \nabla_{\theta_0} \log \pi_{\theta_0}\big(a_w^{(i)} \mid s^{(i)}\big), \qquad A_{\theta_0}[:, 2i] = \nabla_{\theta_0} \log \pi_{\theta_0}\big(a_l^{(i)} \mid s^{(i)}\big).$$

The AuxDPO margin augments DPO with $\delta$:

$$m_i(\theta, \delta) = \beta\bigg( \underbrace{\log \pi_\theta\big(a_w^{(i)} \mid s^{(i)}\big) - \log \pi_\theta\big(a_l^{(i)} \mid s^{(i)}\big)}_{\text{model}} - \underbrace{\Big[\log \pi_{\theta_0}\big(a_w^{(i)} \mid s^{(i)}\big) - \log \pi_{\theta_0}\big(a_l^{(i)} \mid s^{(i)}\big)\Big]}_{\text{reference}} + \underbrace{\delta_{2i-1} - \delta_{2i}}_{\text{Aux term}} \bigg).$$

The training objective is the usual DPO logistic loss $\sum_{i=1}^n \log \sigma\big(m_i(\theta, \delta)\big)$, with constraints/penalties described below.

**Null-space formulation (small $d$; e.g., LoRA/PEFT).** When $d$ is modest, we enforce $A_{\theta_0}\delta = 0$ exactly by optimizing in $\mathcal{N}(A_{\theta_0})$. Compute an orthonormal basis $\Gamma = [\gamma_1, \ldots, \gamma_{2n-r}] \in \mathbb{R}^{2n \times (2n-r)}$ for $\mathcal{N}(A_{\theta_0})$ (e.g., thin SVD/Householder) and parameterize

$$\delta = \Gamma c, \qquad c \in \mathbb{R}^{2n-r}.$$

This guarantees the constraint by construction and reduces variables; we then optimize $(\theta, c)$ jointly with the same optimizer/schedule as DPO.

**Batchwise relaxation (large $d$).** For large models where global $A_{\theta_0}$ is infeasible, we enforce the constraint approximately per batch $\mathcal{B} \subset \{1, \ldots, n\}$ (with $|\mathcal{B}| = B$). Let $A_{\theta_0, \mathcal{B}}$ and $\delta_{\mathcal{B}}$ denote the batch columns/entries. We add a soft penalty and a small stabilizer:

$$\mathcal{L}_{\text{AuxDPO}} = \sum_{i \in \mathcal{B}} \log \sigma\big(m_i(\theta, \delta)\big) - \lambda_{\text{null}}\big\|A_{\theta_0, \mathcal{B}}\delta_{\mathcal{B}}\big\|_2^2 + \lambda_{\text{amp}}\big\|\delta_{\mathcal{B}}\big\|_2^2,$$

and maintain $\|\delta_{\mathcal{B}}\|_2 > 0$ (e.g., via normalization or a small floor) to avoid the trivial solution. This aligns $\delta$ with $\mathcal{N}(A_{\theta_0})$ *locally* while avoiding full-matrix costs.

**Integration details.** We implement AuxDPO as a lightweight extension of `TRL`'s `DPOTrainer`. A custom collator attaches the dataset index $i$ so the trainer can address $\delta_{2i-1}$ and $\delta_{2i}$. Reference log-probabilities are computed once per batch with prompt tokens masked (gradients over response tokens only) to build $A_{\theta_0, \mathcal{B}}$ on the fly in the large-$d$ regime; in the small-$d$ regime, $A_{\theta_0}$ can be precomputed at initialization. We store and update either the per-example vector $\delta$ directly or the lower-dimensional coefficients $c$ (null-space case). All other training hyperparameters (token budgets, splits, early stopping) match standard DPO to ensure comparability.

**What is stored.** *(i)* The per-example offsets $\delta \in \mathbb{R}^{2n}$ (or coefficients $c$ in the null-space parameterization). *(ii)* Either the global $A_{\theta_0}$ (small $d$) or batchwise slices $A_{\theta_0, \mathcal{B}}$ (large $d$). Both are kept in the same precision as the model logits to minimize overhead.

**The hyperparameter $\lambda$.** In our experiments, we vary the hyperparameter $\lambda$ in a logarithmic grid $(10, 1, 0.1, 0.01)$, and find that $\lambda = 1$ achieves the highest accuracy for both RewardBench-V2 and MMLU-pro benchmarks. The results reported in the paper are for $\lambda = 1$. Table 3 reports AuxDPO's sensitivity with $\lambda$ on the MMLU-Pro benchmark under the in-domain (ID) setting using Llama3.2-1B model:

Table 3: Performance of AuxDPO for different values of the hyperparameter $\lambda$.

| AuxDPO Hyperparameter $\lambda$ | 10 | 1 | 0.1 | 0.01 |
|---|---|---|---|---|
| % improvement over base policy | 44.62 | **45.83** | 42.21 | 39.27 |

Table 4: Implementation details for DPO / AuxDPO with an 8B policy. Code is available at https://github.com/Debangshu93/AuxDPO

| Aspect | Details |
|---|---|
| **Trainer & Args** | • `AuxDPOTrainer` (TRL); baseline: `DPOTrainer`.
• Key `DPOConfig`: `per_device_train_batch_size=4`, `num_train_epochs=1`, `eval_strategy="steps"`, `eval_steps=500`, `logging_strategy="steps"`, `logging_steps=100`, `remove_unused_columns=False`. |
| **Precision / Optimizer** | • Trainable params $\approx 8.03$B; dtype `torch.float32` for weights.
• AdamW in FP32 (moments $m, v$ FP32). Runtime `bf16=True` for compute (autocast). |
| **AuxDPO knobs** | • $\lambda_{\text{null}} = 1.0$ (penalty on $\|A^\top \delta\|^2$).
• $\lambda_{\text{amp}} = 0.01$ (small negative L2 on batch $\delta$).
• `delta_cap` $= 1.0$ (tanh bound on $|\delta|$).
• `aux_lr` $= 5 \times 10^{-3}$. |
| **DPOP knobs** | • $\lambda_{\text{pos}} = 1.0$ (penalty encouraging higher log-likelihood on chosen responses). |
| **Reference policy** | • `ref_model=None` (TRL snapshots and freezes a copy).
• Used forward-only for the reference model passes. |
| **Parameter / state sizes (theoretical)** | • Weights $\approx 29.9$ GiB ($8.03$B$\times 4$ B).
• Gradients $\approx 29.9$ GiB.
• Adam moments $(m, v) \approx 59.8$ GiB.
• Total model states $\approx 119.6$ GiB per full replica (excl. activations). |
| **Hardware / GPUs** | • $8\times$ GPUs, $\approx 192$ GiB VRAM each (ROCm total 206,141,652,992 B).
• Single Python process with contexts on all 8 devices. |
| **VRAM (AuxDPO)** | • `allocated`: 17.4 GiB on GPU1–6; 11.9 GiB on GPU0,7.
• `reserved`: 81.2 GiB on GPU1–6; 54–55 GiB on GPU0,7.
• Device-used (ROCm SMI): 82.4 GiB on GPU1–6; 55–56 GiB on GPU0,7. |

| Aspect | Details |
|---|---|
| **VRAM (DPO)** | • `allocated`: 16.25 GiB on GPU1–6; 11.09 GiB on GPU0,7.
• `reserved`: 25.94 GiB on GPU1–6; 20.24 GiB on GPU0,7.
• Device-used (ROCm SMI): 29.4 GiB on GPU1–6; 22–23 GiB on GPU0,7. |
| **Memory instrumentation** | • Step callback prints `allocated`, `reserved`, per-step peaks via `torch.cuda.{memory_allocated, memory_reserved, max_memory_reserved}`.
• Device totals via `rocm-smi -showmeminfo vram`. |

### B.3 DATASET DESCRIPTION

**MMLU-Pro.** MMLU-PRO (Wang et al., 2024b) is a strengthened variant of the Massive Multitask Language Understanding benchmark, designed to stress-test reasoning in large language models. It contains approximately 12,000 multiple-choice questions across 14 subjects (e.g., mathematics, law, chemistry, business, history, psychology). Each question offers 10 candidate answers with a single correct label—expanding the option set from four to ten—to curb guessing and sharpen separation among models. Compared with the original MMLU, prior work reports substantially lower accuracies and reduced sensitivity to prompt style, making MMLU-PRO a robust, reasoning-centric evaluation suite.

**RewardBench v2.** REWARDBENCH V2 (Malik et al., 2025) is a second-generation, multi-skill benchmark for accuracy-based evaluation of reward models on unseen human data. It comprises $\sim 1,870$ prompts spanning six subsets—*Factuality*, *Precise Instruction Following*, *Math*, *Safety*, *Focus*, and *Ties*. Each prompt includes a preferred ("chosen") response and multiple rejected responses. Accuracy is measured by whether the reward model assigns a higher score to the chosen response than to all rejected alternatives. Compared to v1, the v2 release emphasizes harder, out-of-distribution prompts and reports per-subset counts to facilitate reproducible evaluation.[3]

**UltraFeedback.** ULTRAFEEDBACK (Cui et al., 2024) is a large human-preference corpus widely used for preference optimization. We adopt the standardized pairwise format (chosen vs. rejected); the public *preference-standard* split contains approximately 340,000 rows, supporting stable optimization and serving as our training source for out-of-distribution experiments. Following common practice for DPO-style training, we use the preprocessed, binarized release of ULTRAFEEDBACK curated by Dong et al. (2024), which has been shown to yield higher-quality reward models (Dong et al., 2024; Wang et al., 2024a; Xiong et al., 2024; Banerjee and Gopalan, 2024).

### B.4 EXPERIMENTAL RESULTS ON THE HH-RLHF DATASET

HH-RLHF (Bai et al., 2022) is a human-preference dataset released by Anthropic for training reward models that encourage assistants to be both helpful and harmless. The corpus contains roughly 160,000 preference comparisons over single-turn dialogue prompts, collected in several tranches (base models, rejection-sampling models, and iterated "online" RLHF models); each example consists of a user prompt together with two model replies and a binary label indicating which reply is preferred ("chosen") over the other ("rejected"). Helpfulness and harmlessness data are provided in separate but schema-compatible subsets, and the dataset has become a standard benchmark for safety-aware reward modeling and RLHF pipelines. In our experiments, we use the public HH-RLHF release in its pairwise (chosen vs. rejected) format as a canonical baseline preference dataset for assistant-style dialogue.

We present a comparison of AuxDPO with DPO, IPO, and DPOP on this dataset using the Llama3.2-1B model (Table 5). The reported values show percentage change in mean accuracy relative to the base policy under in-domain (ID) and out-of-domain (OOD) settings. The best gains are in bold and the second-best are in italicized fonts. Accuracies that degrade from the base policy are marked in negative. The results show a clear advantage of AuxDPO over the other methods.

---

[3]See the dataset card and paper for construction details, category counts, and scoring.

Table 5: Performance comparison across preference optimization methods over the Anthropic HH-RLHF dataset.

| Method | DPO | AuxDPO | IPO | DPOP |
|--------|-----|--------|-----|------|
| ID | 17.92 | **25.01** | *23.70* | 20.12 |
| OOD | -1.19 | **13.06** | *1.03* | 0.24 |

## B.5 SYNTHETIC EXPERIMENTS

We demonstrate the failure of DPO using the example of Proposition 3. Recall that there are 3 responses with true rewards $r^* = [1, 2, 0]$ and preference ordering $a_2 \succ a_1 \succ a_3$. We take the policy $\pi_\theta \propto [e^\theta, e^{-\theta}, 1]$. The base policy with $\theta_0 = 0$ has the average reward $\pi_{\theta_0}^\top r^*$.

| Method | $\theta$ | $\pi_\theta$ | $\pi_\theta^\top r^*$ |
|--------|----------|--------------|-----------------------|
| DPO | 0.40 | $[0.47, 0.21, 0.32]$ | 0.895 |
| IPO | 0.10 | $[0.37, 0.30, 0.33]$ | 0.969 |
| DPOP | 0.10 | $[0.37, 0.30, 0.33]$ | 0.969 |
| AuxDPO | $-0.50$ | $[0.19, 0.51, 0.30]$ | 1.199 |

Table 6: Comparison of trained policies under an imbalanced pairwise-preference regime ($n_{12} = 5$, $n_{23} = 5$, $n_{31} = 50$). Columns report the learned scalar parameter $\theta$, the policy $\pi_\theta$, and the achieved objective $\pi_\theta^\top r^*$ (higher is better).

| Method | $\theta$ | $\pi_\theta$ | $\pi_\theta^\top r^*$ |
|--------|----------|--------------|-----------------------|
| DPO | $-0.43$ | $[0.21, 0.48, 0.31]$ | 1.17 |
| IPO | $-0.10$ | $[0.30, 0.37, 0.33]$ | 1.03 |
| AuxDPO | $-0.50$ | $[0.19, 0.51, 0.30]$ | 1.20 |

Table 7: Comparison of trained policies under a balanced pairwise-preference regime ($n_{12} = 10$, $n_{13} = 10$, $n_{23} = 10$). Reported are the learned $\theta$, the resulting policy $\pi_\theta$, and the objective $\pi_\theta^\top r^*$ (higher is better).

First, we take pairwise preferences counts $n_{12} = 5$, $n_{23} = 5$ and $n_{31} = 50$. We tabulate the post-optimized policy and its average reward in Table 6. We compare AuxDPO with DPO, IPO, and DPOP. We see that policies output by DPO, IPO, and DPOP change the preference order to $a_1 \succ a_3 \succ a_2$, while AuxDPO is able to maintain the correct ordering. Moreover, the average reward of the AuxDPO policy increases compared to the base policy, whereas for others, the reward decreases, showing their failures. Next, in Table 7, we demonstrate the sensitivity of DPO with respect to pairwise preference counts. For preference counts $n_{12} = 10, n_{23} = 10, n_{31} = 10$,, we see that DPO (as well as IPO) does not suffer from the failure modes and is able to increase the average reward compared to the base policy. Note here that AuxDPO does better than both DPO and IPO.

The results in both Tables together show that DPO is vulnerable to failure modes, including preference reversal, reward reduction, and is sensitive to the relative frequency of pairwise preference counts, while AuXDPO is able to overcome all these.

## B.6 EXAMPLE: BALANCED PREFERENCE DATA FREQUENCIES CANNOT PREVENT REWARD REDUCTION AND PREFERENCE REVERSAL

Recall that the DPO example of Proposition 3 has true rewards $r^* = [1, 2, 0]$, preference pair sampling frequencies in the ratio $n_{1,2} : n_{2,3} : n_{1,3} = 1 : 1 : 10$, and base policy parameter $\theta_0 = 0$. It exhibits the following properties:

- Average reward reduction: The average reward goes down from $1.0$ to $0.895$.

- Likelihood displacement among responses 1 & 2: (Suboptimal) Response 1's probability moves up (from $0.33$ to $0.47$), and (Optimal) Response 2's probability moves down (from $0.33$ to $0.21$).

- Preference reversal among responses 1 & 2: (Suboptimal) Response 1's probability is above (Optimal) Response 2's probability ($0.47 > 0.21$).

By changing, in this example, the pairwise data proportions to uniform ($n_{1,2} : n_{2,3} : n_{1,3} = 1 : 1 : 1$), DPO gives:

- No average reward reduction: The average reward goes up from $1.0$ to $1.17$.

- No likelihood displacement or preference reversal among responses 1 & 2: Response 1's probability moves down (from $0.33$ to $0.21$), and Response 2's probability moves up ($0.33$ to $0.48$).

However, if we further change the base (reference) policy that DPO takes as input (i.e., pair sampling frequencies $n_{1,2} : n_{2,3} : n_{1,3} = 1 : 1 : 1$, base policy parameter $\theta_0 = 2$), we get the following interesting behavior after DPO:

- Average reward reduction: The average reward goes down from $0.89$ to $0.86$.
- No likelihood displacement or preference reversal among responses 1 & 2: Response 1's probability moves down (from $0.87$ to $0.80$), Response 2's probability moves up (from $0.02$ to $0.03$), and their relative ordering of policy probabilities stays unchanged.
- Likelihood displacement among responses 1 & 3: If we consider only responses 1 & 3, response 1 is stronger than response 3 since it has a better true reward ($1 > 0$). However, its probability falls after DPO (from $0.87$ to $0.80$) while the weaker response 3's probability rises (from $0.12$ to $0.17$).

Thus, even with balanced pairwise data, the interaction between the policy class, the reference policy, and the misspecified reward model can produce undesirable outcomes, such as a reduction in average policy reward and relative likelihood displacement.

In general, the only way to choose sampling frequencies that are 'reliable' or 'compatible for DPO' is via exploiting the misspecification geometry via the knowledge of $r^*$, which is impossible. We also remark that the preference datasets used in our experiments (like all other publicly available preference datasets) are inherently imbalanced – every response occurring in the dataset is compared only with one other response, and not with any other occurring response. Moreover, there are many responses (strings) that are never compared.