# OpenReview forum: "Why DPO is a Misspecified Estimator and How to Fix It"
_ICLR.cc/2026/Conference — ICLR 2026 Oral_

### Official Review · Reviewer_egJi · 2025-10-31

**Soundness:** 4
**Presentation:** 2
**Contribution:** 3
**Rating:** 6
**Confidence:** 3

**Summary:**

The paper studies the misspecification of the parametric reward model in the DPO framework.
The standard DPO framework assumes that the LLM is expressive enough to be able to fit the ideal solution characterized by RLHF.
The authors raise critical questions that the LLM models can be misspecified with a concrete example (Section 3.1). This example shows that such misspecification may lead to undesirable behavior.
Based on the local analysis of the geometry, they propose to add additional optimization variables which allow enough slackness to the model to fit to the desired RLHF solution.
The experiment results show clear advantage of the proposed method AuxDPO.

**Strengths:**

This paper proposes an interesting idea to further improve the quality of DPO, based on principled local information geometry.
The proposed technique is not only principled but also effective in practice.

**Weaknesses:**

The only downside of the paper is that it is very notation heavy and not easy to follow in the first read.

**Questions:**

- I cannot understand the implication of Remark 4. Do the authors want to argue that Song et al. (2024)'s argument is wrong? If the authors can end up with a different conclusion, what is the difference? I believe that the difference comes from the misspecification, but I appreciate if the authors can clarify.
- I have a question about the notation in Section 4. The authors use $\mathcal{N}(A)\subset\mathbb{R}^m$ to denote the nullspace of $A$. Given this, I do not know how to parse $\\|\\mathcal{N}(A)\\delta\\|^2$, and how it can manifest as in the final objective function. I think this paragraph deserves a further expansion.

#### **Suggestions**
- The notation for dataset is a bit weird. For example, in line 138, consider $\mathcal{D}=\\{(\text{tuple}\_i)\\}_{i=1}^n$.
- The shorthand b/w and w.r.t. read weird. Please expand them in the final version.

---

Overall, I find the core idea compelling and the paper a solid contribution. However, a thorough revision to streamline and simplify the notation would greatly improve the manuscript’s readability and its accessibility to a broader audience.

---

> ### Author Response · Authors · 2025-11-20
>
> > **I cannot understand the implication of Remark 4. Do the authors want to argue that Song et al. (2024)'s argument is wrong? If the authors reach a different conclusion, what is the difference? I believe that the difference comes from the misspecification, but I would appreciate it if the authors could clarify**
>
> Thank you for bringing up this point. We realize that the current Remark 4 is not phrased in the best possible manner, and regret any confusion or misunderstanding that this may have caused.
>
> To clarify, we do not see any problem with the argument of Song et al (2024). Remark 4 is intended to convey the following: Consider the following three statements:
>
> **(S1)** DPO converges to a (near) optimal policy $\theta^*$,
>
> **(S2)** The global coverage condition $\max_{s, a} \frac{\pi_\theta(a\mid s)}{\pi_{\theta_0}(a \mid s)} \leq C$ holds,
>
> **(S3)** $\pi_{\theta_0}(a|s) \geq 1/C$.
>
> Song et al show that (S1) implies (S2), and that (S3) implies (S2). In the case of the 3-response example in our paper, property (S3) holds (with $C=3$), and thus (S2) holds, but (S1) does not hold. This is because (S2) is not sufficient for (S1), and the reason behind this is the misspecification of the reward function relative to the implicit rewards expressed by the policy class. In summary, global coverage is necessary for DPO to converge (as shown by Song et al.), but it is not sufficient (as illustrated in our example).
>
> We will revise the phrasing of Remark 4 if the above explanation is clearer.
>
> > **...in Section 4. The authors use $\mathcal{N}(A) \subset \mathbb{R}^m$ to denote the nullspace of $A$. Given this, I do not know how to parse, nor how it can manifest in the final objective function. I think this paragraph deserves a further expansion**
>
> This is a typo -- we meant to write $||A\delta ||$ instead of $|| \mathcal{N}(A)\delta ||$. Thanks for catching this.
>
> > **Additional Suggestions**
>
> Thank you for your suggestions and time spent in reading the manuscript. We will rectify these notational inaccuracies.

---

> > ### Author Response · Authors · 2025-11-27
> >
> > Dear Reviewer,
> >
> > We sincerely thank you again for your valuable feedback, which has helped us improve the quality of our work. As the discussion deadline is approaching, we kindly request you to review our rebuttal. We have also uploaded a revised PDF of the paper and its appendix, addressing your concerns to the best of our ability. We are happy to address any remaining concerns. Thank you again for your time and effort!
> >
> > Regards,
> >
> > Authors

---

### Official Review · Reviewer_7Hcd · 2025-10-31

**Soundness:** 3
**Presentation:** 3
**Contribution:** 3
**Rating:** 8
**Confidence:** 2

**Summary:**

This paper first demonstrates that DPO implicitly projects the weights of the optimal solution to the manifold of possible reward functions under the given policy class. The authors prove that this can lead to mis-specifications where the reward function does not actually obey preferences.

**Strengths:**

* the paper is mathematically rigorous in demonstrating its claims.
* the work does a good job demonstrating why the mis-specification is a problem through a useful example and follow-up points.
* the results demonstrate strong performance in both in distribution and out of distribution settings.

**Weaknesses:**

* the derivation of the aux DPO objective makes sense, but is justified using a local approximation. While this makes sense, it could be worth pointing out the the AuxDPO solution (at least to my understanding) holds under these approximations only.
* The paper is a bit hard to follow at times as it is particularly dense. I think at various points in the manuscript having more motivation and explanation would be helpful. Why do we want to do a first order approx? What is the meaning of the A matrix?
* In a similar manner, the actual instantiation of AuxDPO is a bit unclear. AuxDPO adds variables in the null space of the A matrix, but how are these variables actually represented in code? Is $\delta$ just predicted as another head of the LLM?
* I understand why the authors used MMLU etc. as a preference learning benchmark, but it (in spirit) does not seem exactly like one since its more of a QA dataset. While DPO can be used in this case I think more benchmarks actually based on human preferences woudl be preferred.

**Questions:**

See weaknesses section.

---

> ### Author Response · Authors · 2025-11-20
>
> > **... it could be worth pointing out that the AuxDPO solution (at least to my understanding) holds under these approximations only**
>
> Please allow us to clarify what we believe is a question about AuxDPO as an algorithm vs. its performance guarantee.
>
> AuxDPO, as such, is an algorithmic procedure that can be run on any dataset, irrespective of any approximation needing to be assumed. The theoretical soundness guarantee of AuxDPO (Proposition 9 in the paper), however, is valid in the local sense, which manifests at sufficiently high $\beta$. We will be happy to add a couple of lines to reflect this before the statement of Proposition 9.
>
> > **I think at various points in the manuscript, having more motivation and explanation would be helpful. Why do we want to do a first-order approximation? What is the meaning of the A matrix?**
>
> Thank you for the suggestion and question.
>
> In our theoretical analysis, the first order approximation is relevant when the KL regularization parameter $\beta$ is sufficiently high, or, in other words, the policy is forced to remain in a local neighborhood of $\theta_0$. This condition is typically enforced in the alignment and policy optimization formulations of RL (e.g., approaches such as Trust Region Policy Optimization (TRPO), Natural Policy Gradient (NPG), etc.).
>
> The matrix $A_{\theta_0}$ is simply the Jacobian matrix of the (non-linear) function $\theta \in \mathbb{R}^d \mapsto r_\theta^\beta(\cdot,\cdot) \in \mathbb{R}^m$. In other words, linearizing this nonlinear function around $\theta_0$ yields the linear transformation $A_{\theta_0}$ .
>
>
> > **In a similar manner, the actual instantiation of AuxDPO is a bit unclear. AuxDPO adds variables in the null space of the A matrix, but how are these variables actually represented in code? Is $\delta$ just predicted as another head of the LLM?**
>
> AuxDPO is ideally supposed to maintain a vector $\delta$ in the null space of the $A$ matrix, of dimension $m$ (total no. of all possible prompt-response pairs). In practice, since $m$ is prohibitively large, as explained in the paper (lines 410-413), we maintain a vector of $2n \ll m$ variables, each corresponding to a prompt and response that occur in the dataset. The ideal constraint $A\delta = 0$ is enforced in an average Monte-Carlo sense over the dataset.
>
> In our experiments, the vector $\delta$ is not the output of any LLM head; rather, it is maintained as a regular PyTorch array.
>
> >  **I understand why the authors used MMLU, etc., as a preference learning benchmark, but it (in spirit) does not seem exactly like one since it's more of a QA dataset. While DPO can be used in this case, I think more benchmarks, actually based on human preferences, would be preferred**
>
> Thanks for your observation about MMLU. Note that Table 1 in the paper presents results on RewardBench, a preference dataset, rather than a QA dataset like MMLU.
>
> Below, we present a comparison of AuxDPO with DPO, IPO, and DPOP on the Anthropic-Helpful-Harmless dataset (also a preference dataset) using the Llama3.2-1B model. The reported values show percentage change in mean accuracy relative to the base policy under in-domain (ID) and out-of-domain (OOD) settings. The best gains are in **bold** and the second-best are *italic* fonts. Accuracies that degrade from the base policy are marked in negative. The results show a clear advantage of AuxDPO over the other methods.
>
> | Method | DPO   | AuxDPO | IPO    | DPOP |
> |--------|-------|--------|--------|-------|
> | ID     | 17.92 | **25.01** | *23.70* | 20.12 |
> | OOD    | -1.19 | **13.06** | *1.03*  | 0.24  |
>
> We will be glad to include this result in the appendix.

---

> > ### Author Response · Authors · 2025-11-27
> >
> > Dear Reviewer,
> >
> > We sincerely thank you again for your valuable feedback, which has helped us improve the quality of our work. As the discussion deadline is approaching, we kindly request you to review our rebuttal. We have also uploaded a revised PDF of the paper and its appendix, addressing your concerns to the best of our ability. We are happy to address any remaining concerns. Thank you again for your time and effort!
> >
> > Regards,
> >
> > Authors

---

> > > ### Comment · Reviewer_7Hcd · 2025-11-27
> > >
> > > Thank you for your detailed response! It addressed a lot of my questions, and I think including some of these points more clearly in the main paper will be helpful to readers. I already thought the paper was good before rebuttal, and will maintain my score for now.

---

### Official Review · Reviewer_zAc8 · 2025-11-01

**Soundness:** 3
**Presentation:** 3
**Contribution:** 3
**Rating:** 6
**Confidence:** 4

**Summary:**

The paper investigates a fundamental limitation of Direct Preference Optimization (DPO), a widely used direct preference alignment method. It shows that when using parametric policy classes (as opposed to the tabular assumption underlying DPO's derivation), DPO suffers from a misspecified statistical estimation problem. This misspecification arises when the true reward function that generates preferences cannot be represented by the chosen policy class, leading to undesirable outcomes. To address this, the authors propose AuxDPO, which augments the DPO loss with auxiliary variables to mitigate misspecification. They demonstrate the empirical effectiveness of AuxDPO on LLM preference alignment tasks.

**Strengths:**

The paper is well written and clearly structured.

Provides an insightful theoretical analysis of DPO and RLHF via Taylor approximation, revealing:
- the local geometry of DPO under parametric policies,
- the local geometry of RLHF optimization, and
- the relationship between RLHF equivalence classes and DPO linearization.

Proposes a novel and principled solution (AuxDPO) to address the identified misspecification issue.

**Weaknesses:**

Could oversampling or undersampling preference pairs to balance frequencies before DPO training mitigate the misspecification issue and yield comparable performance?

There is no discussion or experimental analysis of AuxDPO's sensitivity to its core hyperparameters ($\lambda$ and $n$). How should these values be chosen in practice?

**Questions:**

(Related to the constructive example in Proposition 3): If preference data are balanced (uniform) across $(s, a, a')$, would DPO still exhibit the preference reversal issue shown in the example?

For the datasets used in the experiments, were the preference frequencies balanced or imbalanced?

---

> ### Author Response · Authors · 2025-11-20
>
> > **Over/undersampling preference pairs. If preference data are balanced (uniform), would DPO still exhibit the preference reversal issue shown in the example?**
>
> Thank you for raising this insightful question.
>
> The true reward function in the example is $r^{\star} = [1, 2, 0]$.
>
> 1) Original DPO example in the paper (pair sampling frequencies $n_{1,2}:n_{2,3}:n_{1,3} = 1:1:10$, base policy parameter $\theta_0 = 0$): Recall that in the original example for DPO (details in Table 4 in the appendix), we see:
>
>    a) Average reward reduction: $1.0$ goes down to $0.895$
>
>    b) Likelihood displacement among responses 1 & 2: (Suboptimal) Response 1's probability moves up ($0.33$ to $0.47$), (Optimal) Response 2's probability moves down ($0.33$ to $0.21$)
>
>    c) Preference reversal among responses 1 & 2: (Suboptimal) Response 1's probability is above (Optimal) Response 2's probability ($0.47 > 0.21$)
>
> 2) Example with balanced pair sampling frequencies (pair sampling frequencies $n_{1,2}:n_{2,3}:n_{1,3} = 1:1:1$, base policy parameter $\theta_0 = 0$): By changing only the pairwise data proportions to uniform, we get:
>
>     a) No average reward reduction: $1.0$ goes up to $1.17$
>
>     b) No likelihood displacement or preference reversal among responses 1 & 2: Response 1's probability moves down ($0.33$ to $0.21$), Response 2's probability moves up ($0.33$ to $0.48$)
>
> However, we now have the following interesting modification:
>
> 3) Example with different base policy and balanced pair sampling frequencies (pair sampling frequencies $n_{1,2}:n_{2,3}:n_{1,3} = 1:1:1$, base policy parameter $\theta_0 = 2$): With a different choice of reference policy, we get:
>
>     a) Average reward reduction: $0.89$ goes down to $0.86$
>
>     b) No likelihood displacement or preference reversal among responses 1 & 2: Response 1's probability moves down ($0.87$ to $0.80$), Response 2's probability moves up ($0.02$ to $0.03$), and their relative ordering w.r.t. probabilities stays unchanged
>
>     c) Likelihood displacement among responses 1 & 3: If we consider only responses 1 & 3, response 1 is stronger than response 3 since it has a better true reward ($1 > 0$). However, its probability falls after DPO ($0.87$ to $0.80$) while the weaker response 3's probability rises ($0.12$ to $0.17$).
>
> Thus, even with balanced pairwise data, the interaction between the policy class, the reference policy, and the misspecified reward model can produce undesirable outcomes, such as a reduction in average policy reward and relative likelihood displacement. We will be glad to include these additional illuminating example settings in the appendix.
>
> In general, the only way to choose sampling frequencies that are "reliable" or "compatible for DPO" is via exploiting the misspecification geometry via the knowledge of $r^*$, which is impossible. We also remark that the preference datasets used in our experiments (like all other publicly available preference datasets) are inherently imbalanced -- every response occurring in the dataset is compared only with one other response, and not with any other occurring response. Moreover, there are many responses (strings) that are never compared.
>
> > **AuxDPO's sensitivity to its core hyperparameters ($\lambda$ and $n$). How are these chosen in practice?**
>
> Thanks for bringing up this important detail. First, we note that $n$ is the preference dataset size, which should really be regarded as an input to the algorithm instead of a hyperparameter. $\lambda$, which enforces the nullspace constraint in AuxDPO, is indeed a hyperparameter. In our experiments, we varied $\lambda$ in a logarithmic grid  (10, 1, 0.1, 0.01) and found that $\lambda=1$ achieves the highest accuracy for both RewardBench-V2 and MMLU-pro benchmarks. The results reported in the paper are for $\lambda = 1$. Below, we report AuxDPO's sensitivity with $\lambda$ on the MMLU-Pro benchmark under the in-domain (ID) setting using Llama3.2-1B model:
>
> | AuxDPO Hyperparameter $\lambda$ | 10 | 1 | 0.1 | 0.01 |
> |--------|-------|--------|--------|-------|
> | % improvement over base policy | 44.62 | **45.83** | 42.21 | 39.27 |
>
> We will be glad to include this sensitivity analysis in the appendix.

---

> > ### Comment · Reviewer_zAc8 · 2025-11-26
> >
> > I thank the authors for addressing all my concerns explicitly. It would be great to add some of these discussions to the appendix/paper as remarks. I will update my score.

---

> > > ### Author Response · Authors · 2025-11-27
> > >
> > > Dear Reviewer,
> > >
> > > We sincerely thank you again for your valuable feedback, which has helped us improve the quality of our work. We have added these discussions to the appendix and uploaded a revised version.
> > >
> > > Regards
> > > Authors

---

### Author Response · Authors · 2025-11-27
**Message to Everyone**

Dear All,

We have uploaded a revised PDF of the paper and its appendix, addressing all concerns raised by the reviewers. We are happy to address any remaining concerns. Thank you again for your time and effort!

Regards
Authors

---

### Meta-Review · Area_Chair_vbU3 · 2025-12-28

**Summary:**

The paper discusses why DPO is not the correct objective when going beyond tabular case and propose a fix as AuxDPO via understanding the local behavior of two-stage RLHF for a parametric class and relating it to a natural gradient step in policy space. The observation is insightful. All the reviewers agree that the paper is novel and interesting. The authors should take into account the comments by the reviewers about simplifying the notation and providing more motivation and intuition.

**Reviewer Concerns:**

All reviewers believed that this is an interesting results. Some reviewers ask about more intuition and explanation which the response addresses in a good way.

**Reviewer Scores:**

Reviewer egJi: Rating: 6
Reviewer 7Hcd: Rating: 8
Reviewer zAc8: Rating: 6 ->8

---

### Decision · Program_Chairs · 2026-01-26

Accept (Oral)